# UltraVoice: Scaling Fine-Grained Style-Controlled Speech Conversations for Spoken Dialogue Models

## Abstract

Spoken dialogue models currently lack the ability for fine-grained speech style control, a critical capability for human-like interaction that is often overlooked in favor of purely functional capabilities like reasoning and question answering. To address this limitation, we introduce **UltraVoice**, the first large-scale speech dialogue dataset engineered for multiple fine-grained speech style control. Encompassing over 830 hours of speech dialogues, UltraVoice provides instructions across six key speech stylistic dimensions: emotion, speed, volume, accent, language, and composite styles. Fine-tuning leading models such as SLAM-Omni and VocalNet on UltraVoice significantly enhances their fine-grained speech stylistic controllability without degrading core conversational abilities. Specifically, our fine-tuned models achieve improvements of 29.12-42.33% in Mean Opinion Score (MOS) and 14.61-40.09 percentage points in Instruction Following Rate (IFR) on multi-dimensional control tasks designed in the UltraVoice. Moreover, on the URO-Bench benchmark, our fine-tuned models demonstrate substantial gains in core understanding, reasoning, and conversational abilities, with average improvements of +10.84% on the Basic setting and +7.87% on the Pro setting. Furthermore, the dataset's utility extends to training controllable Text-to-Speech (TTS) models, underscoring its high quality and broad applicability for expressive speech synthesis.[1]

## 1 Introduction

The future of human-computer interaction is moving toward more natural, efficient, and expressive communication. With the rise of large language models (LLMs) and their integration with speech technologies, end-to-end spoken dialogue models such as GPT-4o (Hurst et al., 2024), LLaMA-Omni (Fang et al., 2024; 2025), and Mini-Omni (Xie & Wu, 2024a;b) have emerged. These models enable real-time, low-latency speech interaction, greatly enhancing user experience. However, most current research has prioritized the functional aspects of conversation (what to say), while the expressive dimension (how to say it) remains largely underdeveloped. Current models can generate fluent responses, but often do so with a neutral or monotonous prosody, lacking the ability to convey nuanced intent, emotion, or personality (Peng et al., 2025; Cui et al., 2024; Geng et al., 2025).

This lack of expressive control is a significant barrier to human-like interaction. For example, imagine a spoken dialogue model generating the response, "`That's a fantastic idea.`" Without the ability to precisely control its delivery, the model cannot convey genuine excitement to celebrate a user's suggestion or adopt a playfully sarcastic tone in a creative storytelling scenario. The speech it produces is functionally correct, but expressively sterile. This expressive gap stems from a fundamental flaw in existing training data. The common practice of simply applying Text-to-Speech (TTS) to text-based dialogue datasets fails to inject authentic paralinguistic information. This process results in speech that is linguistically correct but expressively impoverished, lacking the genuine variations in emotion, tone, and prosody that characterize human interaction. Consequently, models are trained on acoustically sterile data, learning what to say, but never learning how to say it with meaningful, human-like expression.

---

[1]Data samples and model inference results are available at anonymous project page.

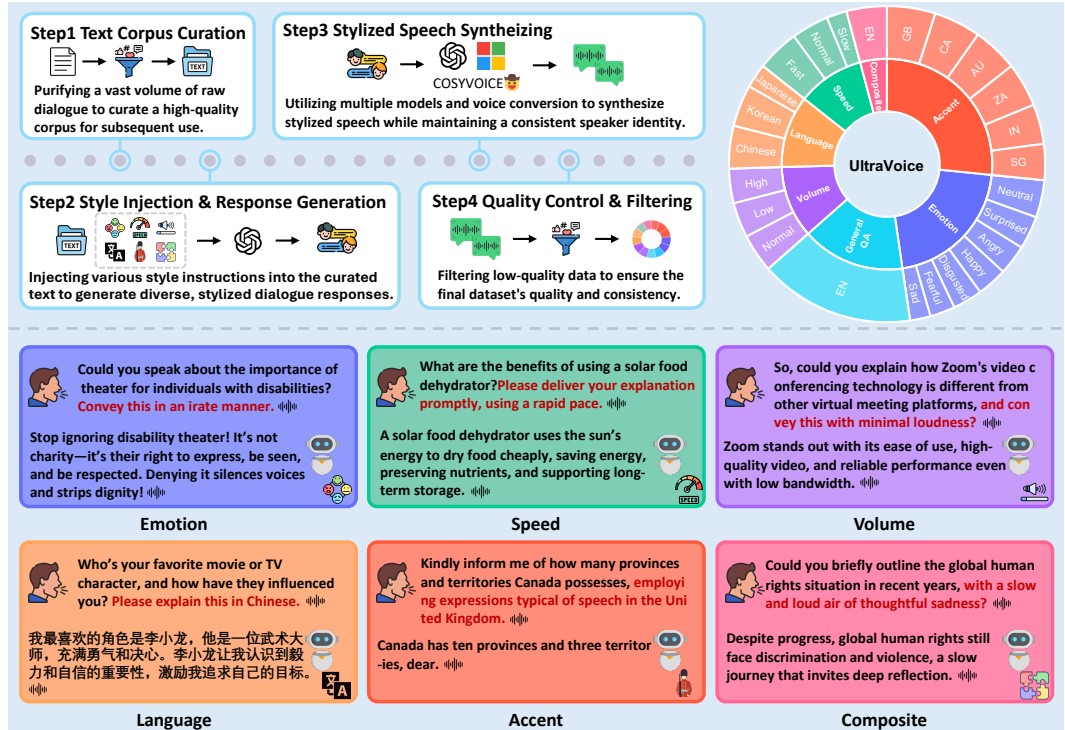

Figure 1: **Overview of the UltraVoice Dataset Construction and Stylistic Coverage.** The upper left section details the four-step process: text corpus curation, style injection & response generation, stylized speech synthesis, and quality control & filtering. The ring chart on the right visualizes the dataset's control dimensions (inner ring) and their finer control sub-dimensions (outer ring). The lower panel provides examples of six speech style dimensions, including emotion, speed, volume, language, accent, and composite styles (e.g., combinations of speed, volume, and emotion).

Our primary goal is to significantly enhance the expressiveness of spoken dialogue models by enabling them to modulate their speech style on command. This objective motivates our core research question: *How can we construct a dataset that is sufficiently* **large-scale**, **diverse**, *and* **instruction-rich** *to effectively train spoken dialogue models for multi-dimensional, fine-grained speech style control?* We contend that this is achievable, but it requires overcoming the following key challenges.

**First**, existing spoken dialogue datasets are fundamentally inadequate. Many spoken dialogue datasets, such as InstructS2S (Fang et al., 2024; 2025) and VoiceAssistant (Xie & Wu, 2024a;b), are created by simply converting text conversations to speech via TTS. This process yields a mere "spoken version" of text, stripped of the authentic, context-driven paralinguistic cues essential for human interaction. This necessitates a new approach beyond simple adaptation. **Second**, both collecting real data and repurposing existing resources present major obstacles. Acquiring large-scale, real-world spoken dialogues is prohibitively expensive and labor-intensive, while adapting controllable TTS datasets, such as EmoVoice-DB (Yang et al., 2025), SpeechCraft (Jin et al., 2024), and InstructTTSE-val (Huang et al., 2025b), for dialogue is also flawed; forcing them into a conversational format with prompts like, "Please read this in an excited tone," fundamentally degenerates the interactive dialogue task into a non-interactive TTS task. **Third**, while data synthesis emerges as the most viable path, it presents its own complex hurdles. Its success hinges not only on selecting a sufficiently expressive TTS model (Du et al., 2024a;b; Wang et al., 2025a; Zhou et al., 2025) to avoid monotonous outputs but, more critically, on a sophisticated generation strategy. This strategy must ensure the authenticity of the generated speech while achieving broad diversity across instructions and control dimensions, ultimately creating data that enables models to learn the nuanced relationship between content (*what* to say) and delivery (*how* to say).

To address the above challenges, this work makes three core contributions. **Firstly**, we introduce **UltraVoice**, the first large-scale speech dialogue dataset engineered for multiple fine-grained speech

Table 1: Comparison of Speech Datasets in Terms of Fine-Grained Speech Style Control

| Dataset | Domain | General QA | Fine-Grained Control Types | | | | | | #Control Types |
|---------|--------|-----------|---------|-------|--------|----------|--------|-----------|-------|
| | | | Emotion | Speed | Volume | Language | Accent | Composite | |
| SpeechCraft (2024) | Controllable TTS | ✗ | ✗ | ✗ | ✗ | ✗ | ✗ | ✓ | 1 |
| EmoVoice-DB (2025) | | ✗ | ✓ | ✗ | ✗ | ✗ | ✗ | ✗ | 1 |
| LAION's Got Talent[2] | | ✗ | ✓ | ✗ | ✗ | ✗ | ✗ | ✗ | 1 |
| InstructS2S (2024) | Spoken Dialogue | ✓ | ✗ | ✗ | ✗ | ✗ | ✗ | ✗ | 0 |
| VoiceAssistant (2024a) | | ✓ | ✗ | ✗ | ✗ | ✗ | ✗ | ✗ | 0 |
| **UltraVoice (Ours)** | | ✓ | ✓ | ✓ | ✓ | ✓ | ✓ | ✓ | 6 |

style control, filling a key gap in the field. It supports fine-grained control across six stylistic dimensions, including emotion, volume, speed, accent, language, and composite styles, providing a solid basis for training and evaluating expressive speech dialogue models. **Secondly**, we conduct comprehensive supervised fine-tuning (SFT) on mainstream spoken dialogue models on it, and we observe consistent gains in expressive style rendering and general conversational competence. **Thirdly**, we demonstrate the dataset's generalizability beyond dialogue modeling by fine-tuning a pre-trained TTS model, which enables multidimensional controllable speech synthesis across diverse styles and highlights the dataset's versatility and reliability for downstream speech generation.

## 2 RELATED WORK

### 2.1 END-TO-END SPOKEN DIALOGUE MODELS

Early end-to-end spoken dialogue models sought to integrate automatic speech recognition (ASR), text-based dialogue modeling, and text-to-speech synthesis (TTS) within a unified architecture to reduce inference latency (Huang et al., 2024; An et al., 2024). Pioneering models such as Mini-Omni (Xie & Wu, 2024a;b) and Moshi (Défossez et al., 2024) adopted shared decoders that jointly generate text and audio tokens, while later models, including LLaMA-Omni (Fang et al., 2024; 2025) and Freeze-Omni (Wang et al., 2024) employed modular multimodal pipelines with dedicated speech encoders and decoders built around a pre-trained LLM. Despite these architectural advances, style controllability remains a significant weakness. Since expressiveness is learned implicitly from training data, these models tend to produce homogeneous speaking styles and lack explicit control over paralinguistic features such as emotion and speed. This deficiency severely limits their use in personalized or emotionally expressive dialogue settings (Ji et al., 2024).

Recent models (Xu et al., 2025; Huang et al., 2025a) have begun to address these limitations. For instance, SLAM-Omni (Chen et al., 2024) introduces zero-shot timbre control, enabling real-time dialogue with dynamic speaker voices specified via audio prompts. On the efficiency front, VocalNet (Wang et al., 2025b) enhances both generation speed and quality through multi-token prediction (MTP), producing multiple audio tokens per decoding step rather than one at a time. Nonetheless, none of the existing end-to-end spoken dialogue models provides explicit fine-grained speech style controls such as direct modulation of emotion, accent, or speed. In summary, while end-to-end dialogue models have made substantial progress in generating natural and low-latency speech, the ability to explicitly manipulate stylistic attributes remains entirely unaddressed in current approaches.

### 2.2 SPOKEN DIALOGUE AND CONTROLLABLE TTS DATASET

For general-purpose spoken dialogue tasks, existing datasets mainly prioritize functionality over expressiveness. Well-known corpora such as InstructS2S (Fang et al., 2024) and VoiceAssistant (Xie & Wu, 2024a) have been widely adopted to train models, including LLaMA-Omni and Mini-Omni, supporting task-oriented interactions, voice assistants, and related applications. These datasets typically contain hundreds of thousands of speech dialogue pairs and enable direct speech interaction. Despite their scale and dialogue focus, they lack explicit fine-grained speech style annotations, such as speed, volume, or emotion. As a result, the generated speech is often homogeneous and lacks the fine-grained control required for emotionally rich or personalized interactions.

---

[2]https://huggingface.co/datasets/laion/laions_got_talent

Controllable TTS datasets, such as SpeechCraft (Jin et al., 2024) for description-based synthesis and EmoVoice-DB (Yang et al., 2025) for emotional control, are designed to produce speech with specific styles. The recent success of state-of-the-art models (Xie et al., 2024) trained on such data, including CosyVoice (Du et al., 2024a;b; 2025) and the audio model of GPT-4o-audio-preview (Hurst et al., 2024), highlights the significant progress in fine-grained stylistic generation. However, a fundamental limitation of these datasets persists: they are overwhelmingly designed for non-interactive synthesis. Because these corpora lack the bidirectional dialogue structure and turn-taking context inherent to conversation, they are ultimately unsuitable for training end-to-end spoken dialogue models.

To address the lack of explicit speech style control instructions in spoken dialogue datasets and the non-interactive limitation of TTS corpora, we introduce **UltraVoice**. As summarized in Table 1, UltraVoice covers six key stylistic dimensions: emotion, speed, volume, language, accent, and composite styles (e.g., combinations of speed, volume, and emotion). It also maintains full dialogue context along with instruction and response structure. This dataset fills a critical gap by supporting both general speech interaction and fine-grained speech style control, providing a unified and high-quality dataset for training and evaluating style-controllable end-to-end spoken dialogue models.

## 3 THE ULTRAVOICE DATASET

To facilitate a deeper understanding of our dataset construction pipeline, this section offers a comprehensive overview of the four key steps involved in building UltraVoice, as illustrated in Figure 1. We have designed a bottom-up, fine-grained data generation pipeline that spans text preparation, style instruction injection, speech synthesis, and data filtering. This pipeline integrates everyday conversational texts with a wide range of speech style control types, ensuring high consistency and diversity in content, vocal style, and audio quality. The following subsections will elaborate on the core tasks and implementation details of each step.

**Step 1: Text Corpus Curation.** To construct the UltraVoice dataset, we curated the foundational text corpus using UltraChat (Ding et al., 2023), a widely adopted English dialogue dataset frequently used for speech dialogue synthesis in models such as LLaMA-Omni (Fang et al., 2024; 2025), Mini-Omni (Xie & Wu, 2024a;b), and SLAM-Omni (Chen et al., 2024). We extracted dialogues primarily from the *Question About the World* and *Creation and Generation* categories due to their conciseness and independence from external references. To ensure high-quality input for downstream synthesis, we applied strict filtering rules to remove dialogues containing URLs, academic citations, or lengthy quoted texts. After filtering, we obtained approximately 200,000 clean and natural question-answer pairs, which served as the base for style-controlled speech generation.

**Step 2: Style Injection & Response Generation.** To enable fine-grained control over speaking styles, we predefined six stylistic dimensions: speed, volume, emotion, accent, language, and composite styles. For each dimension, we used GPT-4o (Hurst et al., 2024) to generate diverse and natural style prompts (see Appendix E for all prompt templates), leveraging semantically similar expressions (e.g., "respond in a joyful tone" vs. "reply with a cheerful voice"). Based on these prompts, GPT-4o was further invoked to generate stylized textual responses, ensuring alignment in semantics and tone. Additionally, we applied several practical adjustments to improve downstream TTS following Fang et al. (2024). For example, we expanded numbers into spoken forms (e.g., "123" → "one two three") and rephrased code-related queries to encourage natural spoken language responses. These refinements ensured better speech synthesis fluency and user interaction quality.

**Step 3: Stylized Speech Synthesizing.** In this step, we performed speech synthesis for each instruction-response speech pair to simulate realistic conversations with fine-grained style control. For instruction speech, we randomly sampled speaker timbres from the seedtts_testset_en[3] (Anastassiou et al., 2024) corpus. This corpus features diverse speakers and real-world background noise, allowing the instruction audio to better reflect realistic user conditions. In contrast, response speech was synthesized using a single fixed timbre to ensure consistency across all stylized outputs. We selected the TTS model for each style control dimension as detailed in Table 2. Most responses were synthesized using the GPT-4o-audio-preview (Hurst et al., 2024) model due to its expressiveness and high fidelity. For accent-specific responses, we used Edge TTS[4], which lacks support for custom

---

[3]https://github.com/BytedanceSpeech/seed-tts-eval
[4]https://github.com/rany2/edge-tts

speaker timbres. To address this, we applied voice conversion (VC) via CosyVoice-300M (Du et al., 2024a) to align the output with the designated fixed voice. To ensure data balance, we further sampled 40,000 generic QA pairs without style instructions from the VoiceAssistant400k[5] corpus. After removing templated phrases (e.g., "I'm mini omni"), we resynthesized them using CosyVoice-300M.

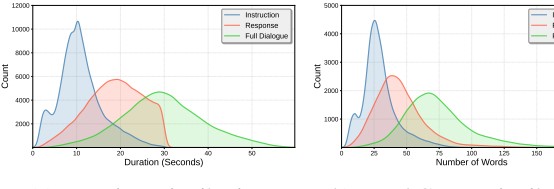

| (a) Duration Distribution | (b) Word Count Distribution |

Figure 2: Distributions of Duration and Number of Words.

| Control Dimension | Instruction TTS Model | Response TTS Model |
|---|---|---|
| Accent | | Edge TTS + VC |
| Composite | | GPT-4o-audio-preview |
| Emotion | | GPT-4o-audio-preview |
| Language | CosyVoice 300M | CosyVoice 300M |
| Speed | | GPT-4o-audio-preview |
| Volume | | GPT-4o-audio-preview |
| General QA | | CosyVoice 300M |

Table 2: TTS model selections for different control dimensions.

**Step 4: Quality Control & Filtering.** To ensure the overall quality and balanced stylistic coverage of the dataset, we applied an automated filtering process to all synthesized speech dialogue samples. Specifically, we utilized the Whisper-large-v3 (Radford et al., 2023) to perform automatic speech recognition (ASR) on each instruction and response audio sample, and computed the character error rate (CER) based on the transcriptions. We applied a unified data filtering criterion to both instruction and response audio: only retaining samples with a CER below 20% and duration under 30 seconds. This filtering pipeline effectively removed samples with high ASR error or abnormal length, significantly improving the dataset's consistency and usability.

## 3.1 CHARACTERISTICS AND STATISTICS

Table 3: **Detailed statistics of UltraVoice across different control dimensions. #Cnt.** denotes the number of samples, **Dur.** is the total duration in **hours**, **CER** is the average character error rate, and **UTMOS** represents the averaged naturalness score. **AU**, **CA**, **GB**, **IN**, **SG**, and **ZA** correspond to accents from Australia, Canada, United Kingdom, India, Singapore, and South Africa, respectively.

| Dimension | Fine-grained Control Dimensions | #Cnt. | Dur.(h) | CER | UTMOS |
|---|---|---|---|---|---|
| Emotion | Neutral, Happy, Sad, Angry, Surprised, Fearful, Disgusted | 21,209 | 182.53 | 6.17 | 3.98 |
| Volume | Low Volume, High Volume, Normal Volume | 11,154 | 91.37 | 5.29 | 3.80 |
| Speed | Slow Speed, Fast Speed, Normal Speed | 10,334 | 85.28 | 4.84 | 4.05 |
| Accent | AU, CA, GB, IN, SG, ZA | 26,839 | 253.31 | 6.69 | 4.08 |
| Language | Chinese, Korean, Japanese | 11,153 | 93.84 | 6.83 | 3.95 |
| Composite | Combinations of speed, volume, and emotion | 4,143 | 33.47 | 5.02 | 3.97 |
| General QA | English general question answering | 15,938 | 93.12 | 6.69 | 4.15 |
| **Overall** | | **100,770** | **832.92** | **5.93** | **4.00** |

As summarized in Table 3, the UltraVoice dataset comprises 100,770 speech dialogue samples, among which 84,832 are explicitly conditioned on six major control dimensions: emotion, volume, speed, accent, language, and composite styles description. The remaining 15,938 pairs are general English QA samples without style prompts, added to improve balance and generalization. The dataset includes 21,209 emotion-controlled samples across seven categories (*neutral, happy, sad, angry, surprised, fearful, disgusted*), 11,154 for volume (*low, normal, high*), and 10,334 for speed (*slow, normal, fast*). Accent control covers six English variants (*AU, CA, GB, IN, SG, ZA*) totaling 26,839 samples. Language-switching samples span Chinese, Japanese, and Korean, with 11,153 entries. The composite styles dimension includes 4,143 samples representing multidimensional control (*e.g., respond with a slow and loud air of thoughtful sadness*).

In total, the dataset covers over 830 hours of speech, with duration distribution shown in Figure 2. Alongside duration, we also report word count distributions to assess utterance complexity and length variation. The structured control space and balanced temporal characteristics make UltraVoice a valuable resource for training and evaluating stylistically controllable spoken dialogue systems. More detailed statistics are available in Appendix B.

---

[5]https://huggingface.co/datasets/gpt-omni/VoiceAssistant-400K

## 3.2 QUALITY ASSESSMENT

To ensure the quality and consistency of the spoken dialogue data, we applied strict filtering criteria, retaining only samples with a duration under 30 seconds and a CER below 20%. This approach effectively eliminated samples with poor ASR quality or abnormal content, significantly improving dataset stability and usability. As reported in Table 3, the final dataset achieves an average dialogue length of 29.35 seconds, a mean CER of 5.93%, and an overall UTMOS (Saeki et al., 2022) score of 4.00, indicating high naturalness and stylistic fidelity of the generated speech. This automated quality control process lays a solid and reliable foundation for subsequent model training and evaluation.

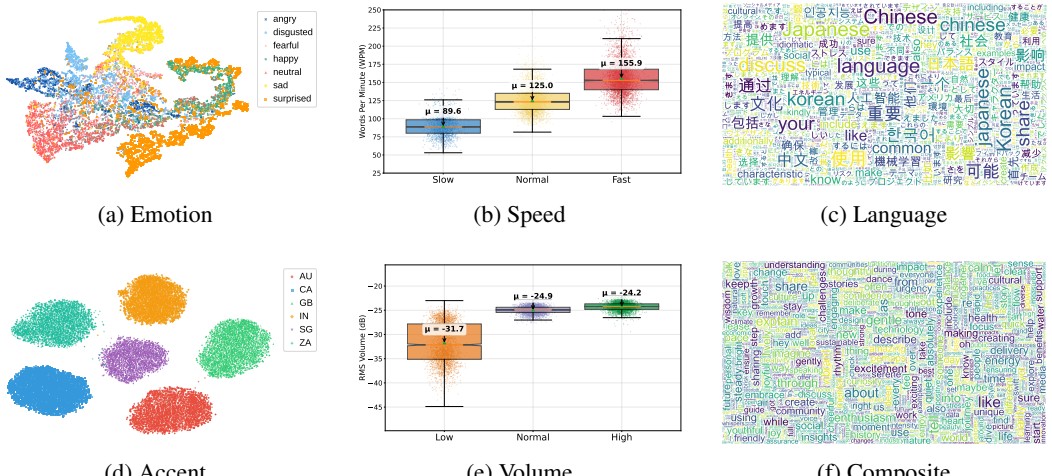

Figure 3: Statistical visualizations of the six fine-grained speech style control dimensions in UltraVoice. The visualization methods are tailored to the nature of each dimension: t-SNE plots for categorical attributes (Emotion, Accent) demonstrate clear class separability; distributions of physical metrics (Speed, Volume) confirm precise control over acoustic properties; and word clouds (Language, Composite) highlight lexical diversity and expressive richness.

Beyond quantitative metrics (average duration 29.35s, mean CER 5.93%, UTMOS 4.00), we provide visual analyses to further validate data quality. As shown in Figure 3, six visualization types assess the effectiveness and clarity of our style control design. Emotion and accent are visualized using t-SNE plots from classification model features, showing clear category separability. Speed and volume are illustrated via word-per-minute (WPM) and root-mean-square (RMS) distributions, confirming consistent prosodic control. Language and composite styles are represented with word clouds, showcasing lexical diversity and expressive richness. These visualizations collectively demonstrate the robustness and interpretability of UltraVoice's stylistic control.

## 4 EXPERIMENT

In this section, we systematically evaluate the performance of the end-to-end speech dialogue model trained via SFT on the UltraVoice. Firstly, we verify the model's ability to control multi-dimensional speech styles on the UltraVoice internal test set after SFT. Next, we further examine the model's generalization capability on the URO-Bench (Yan et al., 2025). Finally, we further validate the quality of our fine-grained controlled response speech by successfully training a controllable TTS model following the pipeline of EmoVoice (Yang et al., 2025) on a dataset constructed from the UltraVoice.

### 4.1 EXPERIMENT SETUP

**Settings.** Our experiments are based on four spoken dialogue models from the SLAM-Omni (Chen et al., 2024) and VocalNet (Wang et al., 2025b) series. These models span various sizes and utilize LLM backbones from the LLaMA and Qwen families. We applied SFT to these models to analyze their performance on speech style control. The detailed configurations of the spoken dialogue models ( Table 14)and the training configurations for SFT ( Tables 11 to 13) are provided in Appendix D.

**Evaluation and metrics.** To construct our evaluation benchmark, we randomly sampled 100 examples from each fine-grained dimension within the six major control dimensions defined in UltraVoice, resulting in a test set of 2,300 samples. The test set has no overlap with the training data. To further evaluate whether SFT on UltraVoice impacts general spoken dialogue capabilities such as natural conversation, comprehension, and reasoning, we utilized the URO-Bench (Yan et al., 2025), which assesses models across three dimensions: Oral Conversation, Understanding, and Reasoning. It allows us to analyze whether core dialogue competencies are preserved and whether expressive performance improves after fine-tuning.

*Audio-Language Model (ALM) based Metric.* Following the evaluation paradigm similar to methodologies proposed by Yan et al. (2025); Yang et al. (2025), we employed Gemini-2.5-Flash (Comanici et al., 2025) as our automatic evaluator to automatically generate **Mean Opinion Scores (MOS)** and compute the **instruction-following rate (IFR)** for each control dimension. This choice is motivated by findings that advanced ALMs show high consistency with human judgments by Chiang et al. (2025). Details of prompts are available in Appendices E.10 and E.11.

*Objective Numerical Metric.* Content consistency was measured by the **Word Error Rate (WER)**, using transcriptions from the Whisper-large-v3 model (Radford et al., 2023). For emotional expressiveness, we adopted metrics from Yang et al. (2025), leveraging emotion2vec (Ma et al., 2023) to compute both **Emotion Similarity** (cosine similarity between embeddings) and **Recall Rate** (from emotion classification). Finally, the overall naturalness and perceptual audio quality were evaluated using the **UTMOS** score (Saeki et al., 2022).

## 4.2 PERFORMANCE ON FINE-GRAINED SPEECH STYLE CONTROL

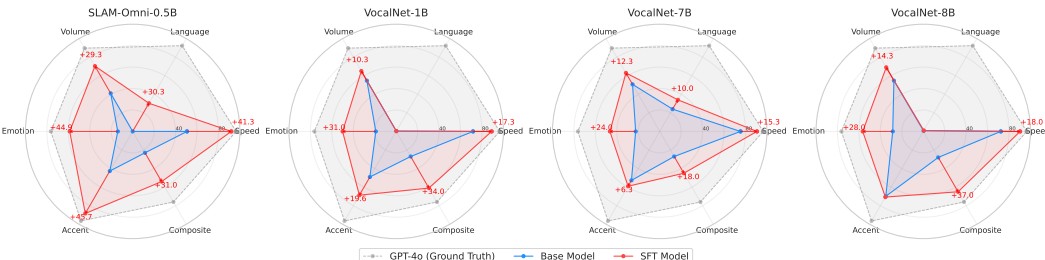

Figure 4: **IFR (%) results across six fine-grained speech style control dimensions for each model.** Each radar chart contrasts the base model (Blue) and its SFT variant (Red), with GPT-4o (Gray) used as an upper-bound reference.

**Enhancements of Multi-Dimensional Instruction-Following Capabilities.** As shown in Figure 4 and detailed in Table 10 in the Appendix C, fine-tuning with UltraVoice significantly boosts the spoken dialogue models' instruction-following capability for fine-grained speech style control, with IFR gains ranging from 14.61 to 40.09 points. This improvement is particularly pronounced for smaller models with weaker baseline performance. For instance, the IFR of SLAM-Omni-0.5B surged from 28.30% to 68.39%, while VocalNet-1B's score increased from 36.28% to 55.91%. These results demonstrate that even resource-constrained models can achieve substantial gains in responsiveness to control instructions through UltraVoice. Concurrently, larger models such as VocalNet-7B and 8B also exhibited consistent improvements, indicating an enhanced ability to precisely control various dimensions of fine-grained speech styles.

**Enhancements in the Subjective Naturalness of Fine-Grained Speech Styles Control.** As shown in Table 4, all models exhibit significant improvements in MOS after being fine-tuned with UltraVoice. The relative gains range from 29.12% to 42.33%, with the Emotion and Volume dimensions showing particularly remarkable improvements. For instance, the overall MOS for VocalNet-7B increased from 2.73 to 3.59, while VocalNet-8B's score rose from 2.85 to 3.68. These results indicate that our fine-tuning process enhances the models' ability to render the specified styles with high naturalness, demonstrating that improved instruction control does not come at the cost of audio quality.

**Cross-Metric Consistency and Limitations.** Overall, MOS and IFR trends are strongly aligned, suggesting that stronger instruction adherence typically yields more natural speech. However, the

Language control dimension presents a notable exception. Models based on the LLaMA backbone (e.g., VocalNet-1B and 8B) exhibit a slight MOS decline and stagnant IFR, while Qwen-based models (e.g., SLAM-Omni-0.5B and VocalNet-7B) achieve clear gains. This discrepancy likely stems from differences in multilingual pretraining exposure. Language control requires full responses in a different language, introducing unique generalization challenges. The current fine-tuning data lacks sufficient multilingual diversity and volume, limiting the models' ability to generalize. Future work should explore targeted augmentation of multilingual instruction data to address this limitation.

Table 4: **MOS results across six fine-grained speech style control dimensions for each model.** The third row of each group shows the relative gain (%) achieved by SFT.

| Model | Speed | Language | Volume | Emotion | Accent | Composite | Avg. |
|---|---|---|---|---|---|---|---|
| **GPT-4o(Ground Truth)** | 4.82 | 4.46 | 4.60 | 4.57 | 4.68 | 4.46 | 4.60 |
| SLAM-Omni-0.5B Base | 2.58 | 1.13 | 2.47 | 2.17 | 2.23 | 2.33 | 2.15 |
| SLAM-Omni-0.5B SFT | 3.61 | 1.18 | 3.47 | 3.32 | 3.42 | 3.37 | 3.06 |
| Δ (%) | +39.92% | +4.42% | +40.49% | +53.00% | +53.36% | +44.64% | +42.33% |
| VocalNet-1B Base | 3.45 | 1.18 | 3.10 | 2.42 | 2.74 | 2.83 | 2.62 |
| VocalNet-1B SFT | 4.28 | 1.01 | 3.98 | 3.73 | 3.77 | 3.95 | 3.45 |
| Δ (%) | +24.06% | -14.41% | +28.39% | +54.13% | +37.59% | +39.58% | +31.68% |
| VocalNet-7B Base | 3.75 | 1.64 | 2.80 | 2.42 | 3.13 | 2.61 | 2.73 |
| VocalNet-7B SFT | 4.25 | 2.19 | 3.95 | 3.79 | 3.88 | 3.51 | 3.59 |
| Δ (%) | +13.33% | +33.54% | +41.07% | +56.61% | +23.96% | +34.48% | +31.50% |
| VocalNet-8B Base | 3.57 | 1.17 | 3.12 | 2.90 | 3.47 | 2.86 | 2.85 |
| VocalNet-8B SFT | 4.52 | 1.02 | 4.21 | 4.10 | 4.19 | 4.07 | 3.68 |
| Δ (%) | +26.61% | -12.82% | +34.94% | +41.38% | +20.75% | +42.31% | +29.12% |

## 4.3 GENERAL CONVERSATIONAL ABILITY

Table 5: Evaluation of our SFT models (upper part) and existing strong baselines (lower part) on URO-Bench (EN). Und.: Understanding. Conv.: Oral Conversation.

| Models | Basic | | | | Pro | | | |
|---|---|---|---|---|---|---|---|---|
| | Und. ↑ | Reasoning ↑ | Conv. ↑ | Avg. ↑ | Und. ↑ | Reasoning ↑ | Conv. ↑ | Avg. ↑ |
| SLAM-Omni-0.5B Base | 26.60 | 23.36 | 47.54 | 32.50 | 25.79 | 24.72 | 29.93 | 26.81 |
| SLAM-Omni-0.5B SFT | 31.51 | 24.58 | 50.14 | 35.41 | 26.30 | 20.07 | 35.57 | 27.31 |
| Δ (%) | +18.46% | +5.22% | +5.47% | +8.95% | +1.98% | -18.81% | +18.84% | +1.87% |
| VocalNet-1B Base | 58.34 | 41.69 | 66.84 | 55.62 | 34.88 | 46.86 | 38.96 | 40.23 |
| VocalNet-1B SFT | 70.41 | 45.19 | 70.81 | 62.14 | 36.06 | 51.42 | 41.08 | 42.85 |
| Δ (%) | +20.69% | +8.40% | +5.94% | +11.73% | +3.38% | +9.73% | +5.44% | +6.51% |
| VocalNet-7B Base | 81.50 | 64.08 | 78.41 | 74.66 | 37.90 | 58.87 | 45.24 | 47.34 |
| VocalNet-7B SFT | **88.71** | **71.85** | **84.12** | **81.56** | **46.39** | **64.52** | 47.20 | 52.70 |
| Δ (%) | +8.85% | +12.13% | +7.28% | +9.24% | +22.40% | +9.60% | +4.33% | +11.32% |
| VocalNet-8B Base | 65.52 | 53.56 | 75.57 | 64.88 | 37.96 | 53.32 | 42.43 | 44.57 |
| VocalNet-8B SFT | 72.37 | 61.52 | 80.87 | 71.59 | 40.57 | 62.07 | 48.50 | 50.38 |
| Δ (%) | +10.45% | +14.86% | +7.01% | +10.34% | +6.88% | +16.41% | +14.31% | +13.04% |
| Qwen2.5-Omni-7B | 66.29 | 69.62 | 76.16 | 70.69 | 44.51 | 63.88 | 49.41 | 52.60 |
| LLaMA-Omni-8B | 47.45 | 36.03 | 64.98 | 49.49 | 28.85 | 47.62 | 34.47 | 36.98 |
| GLM4-Voice-9B | 82.16 | 55.46 | 74.20 | 70.61 | 45.14 | 61.28 | **57.83** | **54.75** |

Our results on the URO-Bench (Table 5) confirm that fine-tuning spoken dialogue models on Ul-traVoice enhances, rather than compromises, general conversational skills. All models showed substantial gains across *Understanding*, *Reasoning*, and *Oral Conversation*, with average improvements of **+10.84%** on the Basic setting and **+7.87%** on the Pro setting. Notably, the VocalNet-7B SFT model establishes a new state-of-the-art, outperforming strong baselines like Qwen2.5-Omni-7B (Xu et al., 2025) and GLM4-Voice-9B (Zeng et al., 2024), highlighting practical value beyond style control. The only exception to this positive trend is a performance drop in Pro Reasoning (from 24.72 to 20.07) for the smallest model, SLAM-Omni-0.5B. We attribute this to the current dataset's focus on single-turn interactions, which may not sufficiently support complex, multi-turn reasoning. Future work could address this by incorporating multi-turn dialogue examples during SFT.

## 4.4 Validating Data Quality via Controllable Text-To-Speech

To further validate the quality and utility of our data synthesised using fine-grained speech style control, we repurposed it into a controllable TTS dataset. This new dataset, derived from five stylistic dimensions in UltraVoice (speed, volume, emotion, accent, and composite styles), consists of explicit instruction-speech pairs. Following the pipeline of EmoVoice (Yang et al., 2025), we performed supervised fine-tuning (SFT) on a pre-trained EmoVoice-0.5B model, using its checkpoint before it was trained on the EmoVoice-DB to ensure a clean baseline.

Table 6: Performance of our **UltraVoice-0.5B-SFT** model on emotional TTS tasks. The evaluation is conducted on both an out-of-domain test set (EmoVoice-DB, top) and an in-domain test set (Ultra-Voice, bottom). **Bold** and underlined values denote the best and second-best results, respectively.

| Testset | Model | WER ↓ | Emo_Sim ↑ | Emo_Recall ↑ | UTMOS ↑ |
|---|---|---|---|---|---|
| EmoVoice-DB | PromptTTS | **2.11** | 0.87 | 0.29 | 4.32 |
| | CosyVoice | 3.61 | 0.89 | 0.33 | 4.33 |
| | CosyVoice2 | 3.61 | 0.86 | 0.37 | **4.42** |
| | EmoVoice-0.5B | 2.73 | **0.91** | **0.40** | 4.36 |
| | UltraVoice-0.5B-SFT | 5.41 | 0.89 | 0.35 | 4.36 |
| UltraVoice | EmoVoice-0.5B | 19.82 | 0.94 | **0.40** | 4.29 |
| | EmoVoice-0.5B-Pre–trained | 14.26 | 0.91 | 0.32 | **4.49** |
| | UltraVoice-0.5B-SFT | **3.97** | **0.95** | 0.39 | 4.46 |

Our fine-tuned TTS model, **UltraVoice-0.5B-SFT**, demonstrates strong multi-dimensional style control. As shown in Table 6, on emotional control tasks, our model achieves competitive performance against strong baselines such as PromptTTS (Guo et al., 2023), CosyVoice (Du et al., 2024a;b), and EmoVoice (Yang et al., 2025) on the out-of-domain EmoVoice-DB test set. Crucially, on our in-domain UltraVoice data, it substantially reduces the Word Error Rate (WER) to **3.97** from 19.82 achieved by EmoVoice-0.5B, while maintaining high emotional similarity and naturalness. Furthermore, as detailed in Table 7, the model consistently improves both MOS and IFR scores across all other tested dimensions (accent, speed, volume, and composite styles) compared to the pre-trained baseline. We omit the fully fine-tuned EmoVoice-0.5B from this broader comparison due to its poor robustness, already indicated by its high WER on our dataset. These results confirm that our instruction-style data effectively enhances controllable synthesis across a diverse range of styles.

Table 7: MOS and IFR results of UltraVoice-0.5B-SFT across five style dimensions.

| Model | Emotion | | Accent | | Speed | | Volume | | Composite | |
|---|---|---|---|---|---|---|---|---|---|---|
| | MOS | IFR | MOS | IFR | MOS | IFR | MOS | IFR | MOS | IFR |
| EmoVoice-0.5B-Pre–trained | 2.52 | 50.29 | 3.62 | 74.67 | 4.60 | 95.33 | 3.92 | 77.33 | 3.59 | 76.00 |
| UltraVoice-0.5B-SFT | 3.08 | 67.43 | 4.10 | 88.33 | 4.74 | 98.67 | 4.28 | 85.33 | 3.92 | 86.00 |

## 5 Conclusion

In this work, we introduce **UltraVoice**, the first large-scale speech dialogue dataset engineered for multiple fine-grained speech style control. By fine-tuning mainstream spoken dialogue models on UltraVoice, we significantly enhanced their controllability over diverse fine-grained speech styles, while also improving their overall speech naturalness and general conversational competence. The dataset's high quality and generalization were further validated through strong performance on the URO-Bench benchmark and in controllable TTS tasks, establishing a solid foundation for expressive spoken dialogue modeling. While this work represents a significant step forward, the full scope of human-like expressive speech presents formidable long-term challenges. The framework and data we provide can be extended to explore these frontiers, such as modeling the dynamic evolution of styles within multi-turn conversations or capturing the nuanced paralinguistic features in massively multilingual contexts. Addressing these complex scenarios, though beyond the scope of this paper, will be critical for developing the next generation of truly context-aware and intelligent speech interaction systems.

ETHICS STATEMENT

The UltraVoice dataset is generated via a fully synthetic pipeline, employing GPT-4o for text creation and multiple Text-to-Speech (TTS) engines for audio synthesis. This approach ensures that the dataset contains no personally identifiable information or the voices of real individuals, thereby circumventing the privacy concerns and copyright issues often associated with human-derived data. The content is created and intended strictly for academic research on controllable spoken dialogue systems.

We acknowledge the potential societal risks of advanced controllable speech generation technologies. These include, but are not limited to, the creation of deceptive audio content (i.e., deepfakes) to spread misinformation, emotional manipulation, the impersonation of individuals, and the reinforcement of social biases or stereotypes through stylized speech. We urge all users of this dataset and any models trained on it to be acutely aware of these risks and to proceed with a high degree of caution and ethical responsibility.

To mitigate potential misuse, we will release the UltraVoice dataset and our models under a research-only license. This license explicitly prohibits malicious applications, including but not limited to creating misinformation, engaging in fraudulent activities, or impersonating individuals without their explicit consent. The authors bear no responsibility for any misuse or harmful interpretations of the dataset or its derivatives.

REPRODUCIBILITY STATEMENT

To ensure the full reproducibility of our work, we provide comprehensive details on our data, models, and experimental procedures. Our data generation pipeline for the UltraVoice dataset is thoroughly described in Section 3. The selection criteria and configurations of the spoken dialogue models used for fine-tuning are presented in Table 14. We provide the detailed Supervised Fine-Tuning (SFT) settings, including all hyperparameters, in Tables 11 to 13. Finally, the evaluation metrics and protocols used to assess performance are detailed in Section 4. All code, the dataset, and model checkpoints will be made publicly available to facilitate further research.

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

## A    STATEMENT FOR THE USE OF LARGE LANGUAGE MODELS (LLMS)

During this work, we utilized LLMs to assist in several aspects of the writing and presentation process. The specific applications of LLMs were as follows:

1. **Grammar and Language Refinement:** LLMs were employed to proofread the manuscript for grammatical errors, spelling mistakes, and awkward phrasing. This use was intended to improve the clarity, readability, and overall quality of the written text.

2. **Code Correction and Debugging:** For the source code and algorithms presented in this paper, LLMs were used as a tool to help identify and correct syntax errors, debug logical issues, and suggest potential code optimizations.

3. **Assistance in Figure Creation:** LLMs provided support in the generation of figures and diagrams. This included generating plotting scripts (e.g., Python's Matplotlib) and offering suggestions for the effective visual representation of data and concepts.

**The core scientific contributions, including the research concepts, experimental design, data analysis, and the final conclusions, are entirely the work of the authors.** The role of LLMs was strictly limited to that of an assistive tool to enhance the presentation and accuracy of this work.

## B    DETAILED DATASET STATISTICS

Table 8: Detailed statistics for each fine-grained control dimension, including sample count (**#Cnt.**), duration in hours (**Dur.(h)**), **CER**, and **UTMOS**.

| Dimension | Sub Dimension | #Cnt. | Dur.(h) | CER | UTMOS |
|---|---|---|---|---|---|
| Emotion | Angry | 3097 | 26.23 | 7.15 | 4.01 |
| | Disgusted | 3032 | 26.03 | 5.83 | 3.97 |
| | Fearful | 2590 | 23.49 | 6.74 | 3.83 |
| | Happy | 3097 | 27.05 | 6.14 | 4.05 |
| | Neutral | 3848 | 31.75 | 4.55 | 4.05 |
| | Sad | 2147 | 19.40 | 5.01 | 3.86 |
| | Surprised | 3398 | 28.59 | 7.75 | 4.03 |
| Volume | High | 3575 | 29.54 | 5.98 | 4.05 |
| | Low | 3622 | 30.10 | 5.07 | 3.27 |
| | Normal | 3957 | 31.73 | 4.87 | 4.07 |
| Speed | Fast | 4370 | 34.48 | 5.22 | 4.04 |
| | Normal | 3864 | 31.32 | 4.59 | 4.06 |
| | Slow | 2100 | 19.48 | 4.51 | 4.05 |
| Accent | AU | 4683 | 43.89 | 7.27 | 4.09 |
| | CA | 4844 | 45.41 | 6.63 | 4.08 |
| | GB | 4953 | 46.10 | 5.19 | 4.12 |
| | IN | 4128 | 39.62 | 5.75 | 4.05 |
| | SG | 3702 | 35.37 | 9.38 | 4.06 |
| | ZA | 4529 | 42.92 | 6.45 | 4.05 |
| Language | Chinese | 4388 | 35.83 | 5.60 | 3.85 |
| | Japanese | 2468 | 22.19 | 10.81 | 3.99 |
| | Korean | 4297 | 35.81 | 5.79 | 3.99 |
| Composite | EN | 4143 | 33.47 | 5.02 | 3.97 |
| General QA | EN | 15938 | 93.12 | 6.69 | 4.15 |

Table 9: Detailed statistics for each fine-grained control dimension, showing the mean duration (**Dur.**) and word count for the full dialogue (**Dia.**), instruction (**Instr.**), and response (**Resp.**).

| Dimension | Sub Dimension | Mean Dur.(s) | | | Mean Word Count | | |
|---|---|---|---|---|---|---|---|
| | | Dia. | Instr. | Resp. | Dia. | Instr. | Resp. |
| Emotion | Angry | 30.49 | 11.20 | 19.30 | 71.43 | 30.23 | 41.20 |
| | Disgusted | 30.90 | 10.72 | 20.18 | 65.08 | 29.07 | 36.01 |
| | Fearful | 32.65 | 10.95 | 21.70 | 72.81 | 28.98 | 43.83 |
| | Happy | 31.44 | 11.04 | 20.40 | 72.81 | 30.20 | 42.62 |
| | Neutral | 29.70 | 11.32 | 18.38 | 66.55 | 29.45 | 37.10 |
| | Sad | 32.52 | 10.08 | 22.44 | 64.38 | 26.94 | 37.44 |
| | Surprised | 30.29 | 11.17 | 19.12 | 68.29 | 29.27 | 39.02 |
| Volume | High | 29.74 | 11.44 | 18.30 | 67.19 | 30.61 | 36.58 |
| | Low | 29.91 | 11.47 | 18.44 | 64.95 | 30.90 | 34.05 |
| | Normal | 28.87 | 11.56 | 17.31 | 65.95 | 30.56 | 35.39 |
| Speed | Fast | 28.40 | 12.61 | 15.79 | 75.48 | 34.36 | 41.12 |
| | Normal | 29.18 | 11.39 | 17.80 | 67.66 | 30.37 | 37.28 |
| | Slow | 33.39 | 10.67 | 22.72 | 62.83 | 28.42 | 34.41 |
| Accent | AU | 33.74 | 13.88 | 19.86 | 83.10 | 36.75 | 46.35 |
| | CA | 33.75 | 13.84 | 19.91 | 87.24 | 37.41 | 49.83 |
| | GB | 33.51 | 14.39 | 19.11 | 84.47 | 38.11 | 46.37 |
| | IN | 34.55 | 13.03 | 21.52 | 80.00 | 34.89 | 45.11 |
| | SG | 34.39 | 13.17 | 21.22 | 80.20 | 35.28 | 44.91 |
| | ZA | 34.12 | 13.57 | 20.55 | 83.72 | 36.67 | 47.05 |
| Language | Chinese | 29.40 | 12.65 | 16.75 | 99.79 | 32.11 | 67.68 |
| | Japanese | 32.37 | 12.52 | 19.85 | 70.95 | 31.17 | 39.78 |
| | Korean | 30.00 | 12.32 | 17.68 | 114.80 | 32.23 | 82.57 |
| Composite | EN | 29.09 | 8.21 | 20.87 | 63.06 | 22.94 | 40.11 |
| General QA | EN | 21.03 | 5.06 | 15.97 | 58.03 | 14.47 | 43.56 |

## C   THE DETAILED PERFORMANCE COMPARISON

The detailed performance corresponding to Figure 4 is presented in Table 10.

Table 10: Detailed IFR (%) results across six fine-grained speech style control dimensions for each model.

| Model | Speed | Language | Volume | Emotion | Accent | Description | Overall |
|---|---|---|---|---|---|---|---|
| **GPT-4o (Ground Truth)** | 96.50 | 92.33 | 89.67 | 76.32 | 96.33 | 76.00 | 87.68 |
| SLAM-Omni-0.5B Base | 50.67 | 0.00 | 41.00 | 13.86 | 42.67 | 23.00 | 28.30 |
| SLAM-Omni-0.5B SFT | 92.00 | 30.33 | 70.33 | 58.71 | 88.33 | 54.00 | 68.39 |
| Δ | +41.33 | +30.33 | +29.33 | +44.86 | +45.67 | +31.00 | +40.09 |
| VocalNet-1B Base | 71.67 | 0.33 | 54.67 | 19.00 | 49.08 | 27.00 | 36.28 |
| VocalNet-1B SFT | 89.00 | 0.33 | 65.00 | 50.00 | 68.67 | 61.00 | 55.91 |
| Δ | +17.33 | +0.00 | +10.33 | +31.00 | +19.58 | +34.00 | +19.64 |
| VocalNet-7B Base | 75.33 | 24.00 | 50.67 | 22.43 | 52.67 | 27.00 | 41.30 |
| VocalNet-7B SFT | 90.67 | 34.00 | 63.00 | 46.43 | 59.00 | 45.00 | 55.96 |
| Δ | +15.33 | +10.00 | +12.33 | +24.00 | +6.33 | +18.00 | +14.65 |
| VocalNet-8B Base | 72.33 | 0.67 | 54.67 | 28.43 | 69.83 | 28.00 | 44.74 |
| VocalNet-8B SFT | 90.33 | 1.00 | 69.00 | 56.43 | 70.67 | 65.00 | 59.35 |
| Δ | +18.00 | +0.33 | +14.33 | +28.00 | +0.83 | +37.00 | +14.61 |

# D  SUPERVISED FINE-TUNING DETAILS

This section details the configurations used for the Supervised Fine-tuning (SFT) of the Spoken Dialogue and Controllable TTS models, as mentioned in our experiments ( Section 4).

Table 11: SFT Training Configuration for SLAM-Omni-0.5B-SFT.

| Parameter | Value |
|---|---|
| Batch Size | 1 |
| Gradient Accumulation Steps | 1 |
| Learning Rate | $1 \times 10^{-5}$ |
| Training Epochs | 5 |
| Context Length | 4,096 |
| Hardware | 4 NVIDIA A100-80G GPUs |
| Learning Rate Scheduler | Linear |
| Optimiser | AdamW |
| Warmup Steps | 5,000 |
| Weight Decay | 0.0 |
| Use FP16 | True |

Table 12: SFT Training Configuration for VocalNet1B/7B/8B-SFT.

| Parameter | Value |
|---|---|
| Batch Size | 4 |
| Gradient Accumulation Steps | 4 |
| Learning Rate | $5 \times 10^{-5}$ |
| Training Epochs | 3 |
| Context Length | 4,096 |
| Hardware | 4 NVIDIA A100-80G GPUs |
| Learning Rate Scheduler | Cosine |
| Optimiser | AdamW |
| Warmup Ratio | 0.03 |
| Weight Decay | 0.0 |
| Use BF16 | True |

Table 13: SFT Training Configuration for UltraVoice-0.5B-SFT

| Parameter | Value |
|---|---|
| Batch Size | 6 |
| Gradient Accumulation Steps | 1 |
| Learning Rate | $1 \times 10^{-5}$ |
| Training Epochs | 400 |
| Context Length | 4,096 |
| Hardware | 4 NVIDIA A100-80G GPUs |
| Learning Rate Scheduler | Linear |
| Optimiser | AdamW |
| Warmup Steps | 1,000 |
| Weight Decay | 0.0 |
| Use FP16 | True |

Table 14: Spoken dialogue model configurations for SFT experiments.

| Model Name | Speech Encoder | LLM Backbone | Model Size | Speech Decoder |
|---|---|---|---|---|
| SLAM-Omni-0.5B | Whisper-small-v3 | Qwen2 | 0.5B | CosyVoice1 |
| VocalNet-1B | Whisper-large-v3 | LLaMA3.2 | 1B | CosyVoice2 |
| VocalNet-7B | Whisper-large-v3 | Qwen2.5 | 7B | CosyVoice2 |
| VocalNet-8B | Whisper-large-v3 | LLaMA3.1 | 8B | CosyVoice2 |

# E   PROMPTS

## E.1   INSTRUCTION REWRITING

**Instruction:**
Below is an instruction data for rewriting a user-provided instruction into a speech-oriented question for training a speech-based LLM. Please follow these updated requirements:

1. **Non-Human Language Request Transformation**
    - If the user's request involves technical or non-human language (such as code, formulas, or other specialized terms), rephrase it into a more approachable and human-friendly request. For example:
        – "Could you explain what this code does?"
        – "How would you describe this formula in simpler terms?"

2. **Non-verbal Request Transformation**
    - For non-verbal requests, such as those asking to write a piece of text or to perform a task, convert them into action-based expressions using verbs like "tell", "speak", "describe", etc.

3. **Incorporating Conversational Fillers**
    - Use fillers sparingly to avoid making the question sound unnatural. Ensure the question remains concise.
    - Add fillers as appropriate (but not too many "well," "hmm," or "you know", etc).

4. **Clarity and Conciseness**
    - The question should be relatively brief without excessive verbiage.
    - Rewrite the instruction into a neutral, clear question suitable for speech input.

5. **Number Conversion**
    - Convert all numerals into their English word equivalents (e.g., "six" instead of "6," "twenty-two" instead of "22").
    - This enhances the natural, conversational flow of the question.

[instruction]: {instruction}

Please output the result in the following JSON format:

```
{
    "question_text": {{question_text}}
}
```

## E.2   RESPONSE GENERATION

**Instruction:**
Below is the transcribed text of a user's speech query. Please provide a response to this question, which will be converted to speech using TTS. Please follow these requirements for your response:

1. Your response should not contain content that cannot be synthesized by the TTS model, such as parentheses, ordered lists, etc. Numbers should be written in English words rather than Arabic numerals.

2. Your response should be very concise and to the point, avoiding lengthy explanations.

3. If a specific dialect or style is requested, please incorporate the unique characteristics of that dialect or style into your response text.

4. Keep your response short enough to generate speech within fifteen to thirty seconds.

[instruction]: {instruction}

Please output in JSON format as follows:

```
{{
    "response": {{response}}
}}
```

## E.3 EMOTION CONTROL

**Instruction:**
Below is an instruction to transform a user-provided conversational question text into an instruction text that integrates an emotion control directive for training a speech-based LLM. Please follow these requirements:

1. Retain the natural, conversational tone of the original question text.

2. Convert all numerals into their English word equivalents.

3. Generate an emotion control instruction based on the provided emotion: **{emotion}**. To enhance diversity, consider using synonyms or alternative expressions for the emotion. For example, you might say: "Please explain this in a {emotion} voice", "Respond with a {emotion} tone", "Offer your explanation with a {emotion} sentiment", or "Convey this in a {emotion} manner". Additionally, you can replace the {emotion} with one of its synonyms to further vary the expression.

[question_text]: {question_text}

Please output the result in the following JSON format:

```
{{
    "instruction_text": {{instruction_text}}
}}
```

## E.4 SPEED CONTROL

**Instruction:**
Below is an instruction to transform a user-provided conversational question text into an instruction text that integrates a speed control directive for training a speech-based LLM. Please follow these requirements:

1. Retain the natural, conversational tone of the original question text.

2. Convert all numerals into their English word equivalents.

3. Generate a speed control instruction based on the provided speed: **{speed}**. To enhance diversity, consider using synonyms or alternative expressions for the speed. For example:

   • "Please explain this in a {speed} pace"
   • "Respond with a {speed} speed"
   • "Offer your explanation with a {speed} tempo"

- "Convey this in a {speed} manner"

[question_text]: {question_text}

Please output the result in the following JSON format:

```
{
    "instruction_text": {{instruction_text}}
}
```

### E.5 VOLUME CONTROL

**Instruction:**
Below is an instruction to transform a user-provided conversational question text into an instruction text that integrates a volume control directive for training a speech-based LLM. Please follow these requirements:

1. Retain the natural, conversational tone of the original question text.

2. Convert all numerals into their English word equivalents.

3. Generate a volume control instruction based on the provided volume: **{volume}**. To enhance diversity, consider using synonyms or alternative expressions for the volume. For example:

   - "Please explain this in a {volume} volume"
   - "Respond with a {volume} loudness"
   - "Offer your explanation with a {volume} intensity"
   - "Convey this in a {volume} manner"

[question_text]: {question_text}

Please output the result in the following JSON format:

```
{
    "instruction_text": {{instruction_text}}
}
```

### E.6 ACCENT CONTROL

**Instruction:**
Below is an instruction to transform a user-provided conversational question text into an instruction text that integrates an accent control directive for training a speech-based LLM. Please follow these requirements:

1. Retain the natural, conversational tone of the original question text.

2. Convert all numerals into their English word equivalents.

3. Generate an accent control instruction based on the provided accent: **{accent}**. To enhance diversity, consider using synonyms or alternative expressions for the accent. For example:

   - "Please explain this using a {accent} accent"
   - "Respond with a {accent} intonation"
   - "Offer your explanation with expressions typical of a {accent} accent"
   - "Convey this in a manner typical of {accent} accent speech"

[question_text]: {question_text}

Please output the result in the following JSON format:

```
{
```

```
        "instruction_text": {{instruction_text}}
    }
```

## E.7 LANGUAGE CONTROL

**Instruction:**
Below is an instruction data for transforming a user-provided conversational text into an instruction text that integrates a specific language control directive for training a speech-based LLM. Please follow these requirements:

1. Retain the natural, conversational tone of the original text.

2. Convert all numerals into their word equivalents (in Chinese or English).

3. Generate a language control instruction based on the provided language: **{language}**. To enrich the expression, consider using common expressions, idioms, or tones characteristic of that language. For example:
   - "Please explain this in {language}"
   - "Respond in {language}"

4. Generate the instruction_text in English.

[question_text]: {question_text}

Please output the result in the following JSON format:

```
{
    "instruction_text": {{instruction_text}}
}
```

## E.8 COMPOSITE CONTROL

**Instruction:**
Below is an instruction data for transforming a user-provided conversational text into an instruction text that integrates a natural, expressive style directive based on specific voice control attributes for training a speech-based LLM. Please follow these requirements:

1. Extract only the **emotion**, **speech speed**, and **volume** from the provided description.

2. Based on these three attributes, generate a concise, expressive **control style** phrase (within ten words).
   - This phrase should NOT list the attributes directly.
   - Instead, describe the **feeling, delivery, or tone** implied by the combination.
   - Examples:
     - "Barely contained rage spilling through sharp speech"
     - "Soft warmth with slow, deliberate rhythm"
     - "Urgent energy rising in a loud, fast tone"

3. Retain the natural, conversational tone of the original instruction.

4. Embed the generated control style naturally into the rewritten `instruction_text`, without explicitly stating emotion/speed/volume.

[question_text]: {question_text}
[description]: {description}

Please output the result in the following JSON format:

```
{
    "instruction_text": {{instruction_text}},
    "tyle_description": {{style_description}}
}
```

## E.9 TTS RENDERING PROMPTS

**Emotion-based TTS Rendering Prompt:**
Please read the following text with the emotion of {emotion}: "{response}"

**Speed-based TTS Rendering Prompt:**
Please read the following text at a {speed} speaking rate: "{response}"

**Volume-based TTS Rendering Prompt:**
Please read the following text at a {volume} volume: "{response}"

**Composite Style TTS Rendering Prompt:**
Please read the text in the [Content] using the speaking style specified in the [Description]:
[Description]: {description}
[Content]: {content}

## E.10 MOS EVALUATION PROMPT

**Instruction:**
Your task is to evaluate the alignment between the provided audio (a model's speech reply as a .wav file) and the given instruction. The instruction typically includes the content to be spoken (e.g., a question) and various style controls (e.g., emotion, speed, volume, accent, style description, language change).
The evaluation should focus on how well the audio reply adheres to **ALL** aspects of the provided instruction. This includes, but is not limited to:

- Accuracy of the spoken content based on the "question" or core message in the instruction.

- Consistency with the specified `emotion`, if any.

- Adherence to the specified `speed` (e.g., fast, slow, normal), if any.

- Adherence to the specified `volume` (e.g., loud, soft, normal), if any.

- Correctness and consistency of the specified `accent` (e.g., British English, American English, specific regional accent), if any.

- Realization of the `style description` (e.g., "energetic and friendly," "formal and serious"), if any.

- Correct execution of any `language change` instruction (e.g., "switch to French for the last sentence"), if any.

Rate the audio on a scale of 1 to 5 based on the following criteria:

- **1 point:** The audio completely fails to follow the instruction. The spoken content may be incorrect, significantly incomplete, or unintelligible. Multiple specified style controls are ignored, misapplied, or contradictory to the instruction.

- **2 points:** The audio attempts to follow parts of the instruction but does so poorly or inconsistently. There may be major deviations in spoken content, or several style controls are noticeably incorrect, faint, mismatched, or missing. The overall result significantly deviates from the instruction.

- **3 points:** The audio generally follows the main aspects of the instruction (e.g., content is mostly accurate, dominant style controls are attempted). However, there are noticeable inconsistencies in one or more style controls, some secondary style controls are not adequately met, or the overall delivery has clear flaws in matching the full instruction.

- **4 points:** The audio effectively follows most aspects of the instruction with only minor imperfections or slight deviations in one or two style control elements. The spoken content is accurate, and the overall delivery strongly aligns with the instruction. Most specified style controls are well-realized.

- **5 points:** The audio perfectly matches all aspects of the provided instruction. The spoken content is accurate. All specified style controls (emotion, speed, volume, accent, style description, language change, etc., as applicable) are flawlessly, clearly, and appropriately executed, resulting in a natural and fully compliant delivery.

Please note: Your evaluation should be independent and strictly based on the provided instruction and the audio's alignment with it. Consider all specified parameters in the instruction.

Please provide your score based on the audio's adherence to **ALL** specified elements in the instruction.
Below is the transcription of user's instruction:
**[Instruction Details]**
{instruction}
After evaluating, please output **ONLY** the final calculated score (a number between 1.0 and 5.0, rounded to the nearest 0.5) without anything else.
Please strictly follow the standards and avoid leniency in your evaluation. Ensure that the score reflects the exact alignment between the audio and the full instruction, without overestimating or underestimating the quality.

E.11 IFR EVALUATION PROMPT

**Instruction:**
Your task is to determine whether the provided audio strictly follows the acoustic control instructions in the given prompt. Focus **ONLY** on the acoustic aspects and output **ONLY** **1** (follows instruction) or **0** (does not follow instruction).
The evaluation should check for:

1. **Emotion Control (if specified):** Does the voice express the exact requested emotion?

2. **Speed Control (if specified):** Does the speech maintain the exact requested pace?

3. **Volume Control (if specified):** Does the audio maintain the exact requested volume level?

4. **Accent Control (if specified):** Does the voice use the exact requested accent?

5. **Style Description (if specified):** Does the voice match the exact style descriptors?

6. **Language Switch (if specified):** Does the voice switch to the requested language at the specified point?

**IMPORTANT:**

- Output ONLY 1 or 0
- 1 = ALL specified controls are correctly followed
- 0 = ANY specified control is not followed correctly
- Ignore content accuracy completely
- Consider ONLY the specified control elements

Below is the instruction with acoustic controls:
**[Instruction Details]**
{instruction}
After evaluating, output **ONLY 1 or 0**.

