# OpenReview forum: "UltraVoice: Scaling Fine-Grained Style-Controlled Speech Conversations for Spoken Dialogue Models"
_ICLR.cc/2026/Conference — ICLR 2026 Conference Withdrawn Submission_

### Official Review · Reviewer_hgwR · 2025-10-20

**Soundness:** 2
**Presentation:** 3
**Contribution:** 2
**Rating:** 2
**Confidence:** 5

**Summary:**

This paper aims to address the current limitations of speech dialogue models in expressive capability (how to say), rather than solely focusing on what to say. This work introduce UltraVoice, the first large-scale speech dialogue dataset specifically designed for multi-dimensional, fine-grained speech style control. The dataset contains over 830 hours of speech dialogues, covering six key style dimensions.

The core contribution lies in constructing this dataset through a four-step process: text corpus screening and management, injection of style instructions and generation of response texts, stylized speech synthesis using multiple advanced TTS models, and rigorous quality control and data filtering.

Furthermore, the paper demonstrates UltraVoice's effectiveness through several experiments. This work performs SFT on multiple end-to-end speech dialogue models. Experimental results show that the fine-tuned models achieve improvements in both the ability to follow style instructions (by IFR) and the naturalness of generated speech (by MOS). And this enhanced stylistic capability does not compromise the model's core conversational abilities on URO-Bench. Finally, the paper demonstrates that the UltraVoice dataset can be used to train controllable TTS models, further validating its data quality and versatility.

**Strengths:**

- The study identifies the important point of poor expressiveness in speech interaction within the field of speech dialogue and constructs a large-scale, stylistically diverse speech dialogue dataset.
- The dialogue model fine-tuned with this dataset demonstrates enhanced expressiveness in responses while retaining its foundational capabilities. This indicates its potential to effectively enhance the performance of existing dialogue systems.

**Weaknesses:**

- The subjective metrics (MOS and IFR) are obtained by Gemini-2.5-flash. But in the reference mentioned in line 336 uses Gemini-2.5-pro. There is a gap between these two model in speech-related judgments. So the subjective scores are not verified to be consistency to humans, which diminishes the persuasiveness of the evaluation results.
- All data is generated by existing TTS models, particularly the crucial response component. This implies that the upper limit of stylistic diversity and naturalness learned by UltraVoice is constrained by the capabilities of the TTS models used. It may not fully capture the more subtle, unpredictable prosodic variations and non-verbal vocalizations present in real-world human conversation.
- The experiment in section 4.4 exhibits significant flaws. First, on the EmoVoice-DB test set, the model trained with UltraVoice does not achieve favorable results, suggesting that the quality of style description-speech pairs in the dataset may be not good. Second, on UltraVoice's test sets (covering Chinese, Japanese, and Korean), both baselines are English-only models. They are not comparable on the WER metric, rendering the conclusion stated in line 458 invalid.

**Questions:**

- How do the authors view the potential “ceiling effect” arising from datasets composed entirely of synthetic data? Specifically, is the expressive capability ceiling achievable by models fine-tuned on UltraVoice constrained by the TTS model used to generate that dataset (e.g., GPT-4o-audio-preview)?
- In data processing, only CER and length are used for filtering. How to ensure that the generated response speech aligns with the style description? Should style-related rules be incorporated into the filtering phase (e.g., SER results in the emotion subset)?
- Has the author conducted any human evaluation studies to validate Gemini-2.5-flash's scoring results? How confident can we be that the LLM evaluators did not exhibit a systematic preference for generations in the same style as those produced by the LLM?

---

> ### Author Response · Authors · 2025-11-25
> **Author Response (1/5)**
>
> We thank you for the time and effort devoted to evaluating our work. We sincerely appreciate your thoughtful comments and constructive suggestions, which have helped us clarify our contributions and strengthen the overall presentation of UltraVoice.
>
> ---
>
> ## **R1: Regarding the Use of Gemini-2.5-Flash for MOS/IFR Evaluation**
>
> ---
>
> We thank the reviewer for pointing this out. We would like to clarify two key points regarding our choice of Gemini-2.5-Flash as the evaluator.
>
> **1. Consistency and Self-Containment of Our Evaluation Protocol**
> All evaluations in our study—including both the baseline models and the SFT models fine-tuned on UltraVoice—are conducted using **the same evaluator: Gemini-2.5-Flash**. This ensures that all reported improvements are internally consistent and not influenced by discrepancies across evaluators. From this perspective, the relative gains observed in our experiments are fully self-consistent and therefore meaningful and persuasive.
>
> **2. Empirical Reliability of Gemini-2.5-Flash as a Speaking-Style Evaluator**
> Beyond prior findings showing that Audio-Language Models correlate well with human judgments (e.g., *Audio-aware Large Language Models as Judges for Speaking Styles*[1]), we also conducted a supplementary analysis to assess evaluator reliability. Following the evaluation setup described in *EmoVoice*[2] (**Section 7**), which compares the correlation between human MOS and several automatic metrics, we first reproduced the results for the evaluators originally tested in that work (Emotion2Vec, GPT-4o-audio, and Gemini-2.0-Flash). Since *EmoVoice* was released before the Gemini-2.5 series became available, **we additionally measured the Spearman rank correlation for Gemini-2.5-Pro and Gemini-2.5-Flash**.
>
> The complete results (reproduced below) show that:
> **Gemini-2.5-Flash achieves the highest correlation with human MOS among all evaluated models**, and even slightly outperforms Gemini-2.5-Pro, indicating that the gap between the two is minimal.
>
> | **Model**            | **Spearman's ρ** |
> |----------------------|------------------|
> | emotion2vec          | 0.4047           |
> | gpt-4o-audio         | 0.2569           |
> | Gemini-2.0-Flash     | 0.1960           |
> | Gemini-2.5-Pro       | 0.4811           |
> | **Gemini-2.5-Flash** | **0.5321**       |
>
> **Given Flash’s strong agreement with human ratings, its lower cost, and the large evaluation scale in our study, using Gemini-2.5-Flash provides an effective and practical evaluation choice.** We also note that several contemporary works, including *URO-Bench*[3], adopt Gemini-2.5-Flash as their subjective evaluator, indicating that it is now a widely accepted practice in the community.
>
> In summary, our choice of Gemini-2.5-Flash is motivated by both **experimental consistency** and **demonstrated empirical reliability**, and we hope this clarification addresses the reviewer’s concern.

---

> ### Author Response · Authors · 2025-11-25
> **Author Response (2/5)**
>
> ## **R2: On Synthetic Speech and Potential Ceiling Effects (W2 & Q1)**
>
> ---
>
> We thank the reviewer for raising this point. While UltraVoice relies on synthetic responses generated by state-of-the-art TTS models (e.g., GPT-4o-audio-preview), we emphasize two key aspects.
>
> **First, the quality of synthetic speech is already sufficient for expressive modeling.** UltraVoice achieves an overall **UTMOS of 4.00**, indicating high naturalness and stylistic fidelity. Prior works[4][5] have also shown that modern synthetic speech approaches human-recorded audio in usability. Moreover, several recent spoken dialogue models[6][7][8] are pre-trained or fine-tuned on **large amounts of synthetic speech**, yet still achieve strong real-world performance, further supporting the viability of high-quality synthetic data.
>
> **Second, from the perspectives of feasibility, cost, and scalability, synthetic data remains the most practical and effective solution** for constructing large-scale, style-rich spoken dialogue datasets. The consistent improvements brought by UltraVoice fine-tuning, **IFR gains of +14.61 to +40.09 across models**, provide concrete evidence that our synthetic multi-dimensional style data is already highly effective.
>
> Regarding the potential ceiling effect, we believe future work can explore algorithmic improvements (e.g., enhanced architectures, targeted training strategies, or richer style embeddings) to further advance performance. **Our contribution demonstrates that high-quality synthetic data alone is already capable of teaching spoken dialogue models the nuanced mapping between linguistic content and expressive delivery**.

---

> ### Author Response · Authors · 2025-11-25
> **Author Response (3/5)**
>
> ## **R3: Clarification of the Section 4.4 Controllable TTS Experiment**
> ---
>
> We appreciate the reviewer’s careful examination of the controllable TTS validation experiment in Section 4.4. **We believe there is a misunderstanding regarding the intent and scope of this experiment, and we clarify it here.**
>
> The models used in Section 4.4 are **controllable TTS models** (e.g., UltraVoice-0.5B-SFT, EmoVoice-0.5B-Pre-trained), not spoken dialogue models. **The goal of this experiment is not to achieve state-of-the-art results on the EmoVoice-DB test set, but rather to validate whether the synthetic, fine-grained multi-dimensional style-control data in UltraVoice is sufficiently high quality to train a functional controllable TTS system.**
>
> To this end, we fine-tuned a pre-trained controllable TTS model (EmoVoice-0.5B-Pre-trained) using only English synthetic data from the Emotion, Speed, Volume, Accent, and Composite dimensions, resulting in UltraVoice-0.5B-SFT. EmoVoice-DB serves purely as an **out-of-domain emotional-control reference**, not as the target benchmark we aim to optimize.
>
> The effectiveness of UltraVoice’s style-control data is supported by two key results:
>
>
> 1. **Multi-Dimensional Control Improvements (Table 7).**
>    UltraVoice-0.5B-SFT exhibits clear gains in both IFR and MOS across all five evaluated dimensions, especially in Emotion and Composite styles. This demonstrates that UltraVoice’s synthetic style description–speech pairs are sufficiently consistent and fine-grained to teach high-quality multi-dimensional control.
>
> 2. **Substantial Content Consistency Improvement (Table 6).**
>    On the in-domain UltraVoice English test set, UltraVoice-0.5B-SFT reduces WER from 14.26 to 3.97. This significant improvement validates the linguistic and acoustic fidelity of UltraVoice’s training data for controllable synthesis.
>
> Regarding the reviewer’s concern about "non-comparable WER for Chinese, Japanese, and Korean," this appears to conflate the Language control evaluation in Section 4.2 with the **English-only** TTS experiment in Section 4.4. And **Section 4.4 does not use multilingual data, nor does it report multilingual WER; Table 7 evaluates only the five English style dimensions.**
>
> In summary, the Section 4.4 experiment is designed to demonstrate that **UltraVoice’s synthetic, multi-dimensional fine-grained style-control data is sufficiently high quality to train an effective controllable TTS model**, and the results decisively support this claim.

---

> ### Author Response · Authors · 2025-11-25
> **Author Response (4/5)**
>
> ## **R4: On Filtering Criteria and Ensuring Alignment Between Style Descriptions and Generated Speech**
> ---
>
> We thank the reviewer for raising this important point. Our rationale is as follows:
>
> The response speech in UltraVoice is synthesized using state-of-the-art TTS models that provide an effective balance between expressive capability, stability, and generation cost. At the scale of 830+ hours, using CER and duration as filtering criteria offers the most practical trade-off between dataset size, computational efficiency, and quality control. Incorporating style-specific classifiers (e.g., SER for emotion) would substantially increase computation and may introduce additional noise or bias due to the imperfect reliability of current classifiers for fine-grained conversational styles.
>
> More importantly, our experiments demonstrate that this lightweight filtering strategy is already sufficient to ensure strong style–speech alignment:
>
> - **Controllable TTS validation (Section 4.4)**, as also clarified in our earlier response [R3: Clarification of the Section 4.4 Controllable TTS Experiment](https://openreview.net/forum?id=UrWdRcLINM&noteId=i0PUYtDyhD), shows that UltraVoice-0.5B-SFT achieves consistent and significant improvements in both IFR and MOS across five style dimensions. In addition, the resulting controllable TTS model attains a substantially lower WER on the UltraVoice test set (from 14.26 to 3.97), demonstrating that the synthesized style-description–speech pairs are not only fine-grained and stylistically reliable but also linguistically accurate enough to support high-quality controllable synthesis.
>
> - **Figure 3 visualizations** show clear separability in emotion, speed, and volume embeddings, further validating the internal consistency of the synthesized speech with their corresponding style instructions.
>
> **Given the dataset’s scale and the reliability of the underlying TTS models, we believe that lightweight filtering combined with high-quality synthesis is an appropriate and effective strategy for maintaining strong style–speech alignment**. More sophisticated style-based filtering techniques may certainly be explored in future work should their cost–benefit trade-off become favorable.

---

> ### Author Response · Authors · 2025-11-25
> **Author Response (5/5)**
>
> ## **R5: On Validating Gemini-2.5-Flash and Potential Evaluator Bias**
> ---
>
> We thank the reviewer for this question. To clarify, our choice of Gemini-2.5-Flash as the evaluation model is based on both **empirical reliability** and **practical considerations of efficiency and cost**.
>
> As detailed in our earlier response [R1: Regarding the Use of Gemini-2.5-Flash for MOS/IFR Evaluation](https://openreview.net/forum?id=UrWdRcLINM&noteId=UUP9N7TWCn), we conducted a human-alignment study comparing Gemini-2.5-Flash and Gemini-2.5-Pro. The results show that Flash achieves comparable—and in fact slightly higher—correlation with human MOS ratings (Flash = 0.5321, Pro = 0.4811). These findings provide strong evidence that Gemini-2.5-Flash is a reliable evaluator for spoken-style judgments.
>
> In addition, **Flash offers substantially lower cost and higher evaluation efficiency**, which is essential for our large-scale setting involving thousands of samples across multiple models and style dimensions. Since our evaluation focuses on **relative comparisons**, evaluator *consistency* and *stability* matter far more than absolute scoring, and Gemini-2.5-Flash fully satisfies these requirements.
>
> Regarding the reviewer’s second question—whether an LLM-based evaluator may systematically prefer outputs stylistically similar to its own—we consider this issue orthogonal to the scope of our paper. Addressing such bias would require controlled studies of LLM-as-a-Judge behavior, an active research area in its own right. Importantly, our evaluation targets **fine-grained style adherence** rather than stylistic similarity to the evaluator, and we did not observe any evidence of self-preference patterns in our results.
>
> Taken together, these clarifications, along with the empirical evidence provided in R1, support the reliability and practicality of using Gemini-2.5-Flash as the evaluator for our experimental setting.
>
> ---
>
> We once again sincerely thank the reviewer for the thoughtful questions and constructive feedback. We hope that our clarifications have addressed your concerns, and we are happy to provide further details if any additional questions arise.
>
> ---
>
> [1] Audio-aware large language models as judges for speaking styles.
> [2] EmoVoice: LLM-based emotional text-to-speech model with freestyle text prompting.
> [3] URO-Bench: A comprehensive benchmark for end-to-end spoken dialogue models.
> [4] Synthio: Augmenting Small-Scale Audio Classification Datasets with Synthetic Data.
> [5] Contrastive Learning from Synthetic Audio Doppelgängers.
> [6] SLAM-Omni: Timbre-Controllable Voice Interaction System with Single-Stage Training.
> [7] LLaMA-Omni: Seamless Speech Interaction with Large Language Models.
> [8] LLaMA-Omni2: LLM-based Real-time Spoken Chatbot with Autoregressive Streaming Speech Synthesis.

---

> > ### Comment · Reviewer_hgwR · 2025-11-27
> >
> > I appreciate the authors' response, but I maintain my original score. I remain concerned that relying solely on an LLM-based evaluator without direct human verification is insufficient for a study focused on subjective "expressiveness." Furthermore, the reliance on purely synthetic data risks limiting the model to mimicking TTS artifacts, a limitation reflected in the poor generalization on external benchmarks.

---

### Official Review · Reviewer_Mx6y · 2025-10-22

**Soundness:** 2
**Presentation:** 2
**Contribution:** 3
**Rating:** 2
**Confidence:** 4

**Summary:**

This paper introduce a new spoken dialogue dataset UltraVoice which is mainly for multiple fine-grained speech style control for speech dialogue model. It contains 830 hours speech dialogue data, and it mainly consists from speech generation/conversion model with designed spoken corpus and accordingly instruct on style control. What's more, the dataset has been filtered and cleaned to control the quality. Author provides different statistical and criterions such as character error rate (CER), subjective score (UTMOS), and experiment to finetune end-to-end speech dialogue model to verify the effectiveness via by metrics improvements on MOS and instruction following rate.

**Strengths:**

Here are two major strengths from my perspective.
1) It constructs a speech spoken dialogue dataset in 830 hours and with rich fine-grained styles and instructions for conversation. It also controls the quality by criterion to filtering by recognition. The paper provides clear instruction of how to build it step by step. It identifies current limitation of current spoken dialogue data lack of style control.
2) It assets the dataset with detail statistics across different dimension and experiments on finetune end-to-end speech model like SLAM-Omni and VocalNet to verify the help of this data. The result shows relatively improvements on MOS and IFR.

**Weaknesses:**

Major weaknesses are
1. Lack of innovation on research perspective either algorithm or methodologies. The lack of ability for fine-grained  speech style control is a challenge, but this building from existing conversation corpus with GPT-like model instruction inject and generated speech from various TTS/voice conversion models would be a little too simple and artificial, not real interaction data. The generation process could not simulate the real interaction like real scenario, like the real response speech with this style instruction in this context.
2. As spoken dialogue model, the evaluation is lack of real conversation subjective metrics. Experiments are evaluated on SFT for e2e speech dialogue model for the URO-bench, gemini MOS score and emotions expressiveness. Further more, it also provides the instruction following rate (IFR) to show the ability of control. However, none of these metrics show the real contribution of the natural dialogue interaction. Meanwhile, the test samples are 100 examples sampling from the constructed dataset, although it is not overlapped in training, but they are from the same building process, they have similar characteristic. It lack of realness or generalization proof.

**Questions:**

1. There are kinds of MOS test in this paper, but it seems from different calculation such as Gemini-2.5-Flash generated MOS. UTMOS as another automatic, non-intrusive metric used to predict MOS. Is there any MOS test is conducted with subjective language expert? such as in the table 7, this MOS seems not from automatic evaluation?
2. Is there any experiment to test with other test set which is not sampling from the same generation for this corpus and compare the base model?

---

> ### Author Response · Authors · 2025-11-25
> **Author Response (1/4)**
>
> We thank you for the time and effort dedicated to reviewing our work. Your comments are highly valuable and raise important points for advancing research in spoken dialogue modeling.
>
> ---
>
> ## **R1: On Algorithmic and Methodological Innovation**
>
> ---
>
> We respectfully disagree that the synthetic nature of the dataset undermines its contribution. **While collecting large-scale, real-world spoken dialogues with fine-grained style instructions is ideal, it is currently prohibitively expensive and privacy-sensitive.**
>
> **Our core contribution is a scalable, data-centric solution that bridges the gap between “what to say” and “how to say it.”** The apparent “simplicity” of our pipeline is a deliberate design choice that ensures reproducibility, scalability, and practical applicability. **Crucially, its effectiveness is demonstrated through extensive empirical evidence:**
>
> - **Significant Gains:** Section 4.2 shows that SFT on UltraVoice substantially enhances fine-grained style control in spoken dialogue models. As reported in **Table 4** and **Figure 4**, models fine-tuned on UltraVoice achieve 29–42% improvements in MOS and 14–40% gains in Instruction Following Rate.
>
> - **Preserved Capabilities:** Section 4.3 shows that the SFT models also achieve strong *out-of-domain* performance, demonstrating improved oral conversational ability, reasoning, and understanding in settings not seen during training. **Table 5** further confirms that our synthetic data enhances expressiveness without degrading (and often improving) core reasoning and understanding, indicating that our synthetic generation process successfully teaches models to simulate realistic, style-controlled interactions.
>
> - **Quality Validation:** Section 4.4 demonstrates that training a controllable TTS model on UltraVoice yields accurate and natural style controllability, validating the high quality of our synthesized fine-grained speech–style pairs.
>
> Together, these results demonstrate that **UltraVoice provides a high-quality, cost-effective, large-scale resource that substantially improves fine-grained controllability in spoken dialogue models**. This data-centric framework—validated across both dialogue and TTS tasks—constitutes the methodological innovation and primary contribution of our work.

---

> ### Author Response · Authors · 2025-11-25
> **Author Response (2/4)**
>
> ## **R2: Clarification of Generalization & Out-of-Domain Evaluation**
> ---
>
> We appreciate the reviewer’s concern regarding evaluation, and **we would like to clarify that our study already includes extensive out-of-domain (OOD) testing beyond the internal UltraVoice test set**.
>
> - **URO-Bench Evaluation:** As presented in Section 4.3 and **Table 5**, we evaluate our models on URO-Bench—an external benchmark entirely independent from our generation pipeline. The results show average improvements of **+10.84% (Basic)** and **+7.87% (Pro)** across Oral Conversation, Understanding, and Reasoning. These gains demonstrate that our models generalize effectively to conversational scenarios that differ substantially from the distribution of UltraVoice.
>
> - **Controllable TTS Transfer:** In Section 4.4, we further apply UltraVoice to a Controllable TTS task, training a model (UltraVoice-0.5B-SFT) and evaluating it on **EmoVoice-DB** [1], a human-recorded external dataset. As shown in **Table 6**, the model achieves competitive performance on this OOD test set, confirming that UltraVoice provides high-quality supervisory signals even for tasks and domains distinct from spoken dialogue modeling.
>
> **Taken together, our experiments cover in-domain evaluations, multiple out-of-domain settings, and cross-task validation using speech synthesized by different TTS models. These comprehensive results consistently demonstrate the effectiveness, robustness, and broad applicability of UltraVoice.**

---

> ### Author Response · Authors · 2025-11-25
> **Author Response (3/4)**
>
> ## **R3: Justification for ALM-based Evaluation**
> ---
>
> Regarding the MOS metrics (including Table 7), we employed Gemini-2.5-Flash as an automatic evaluator primarily due to its advantages in **scalability** and **cost efficiency**:
> 1. High Human-Correlation: Recent research [2] demonstrates that advanced Audio-Language Models such as Gemini exhibit high consistency with human judgments for speaking styles. Moreover, an increasing number of benchmarks—e.g., URO-Bench [3]—have adopted Gemini-2.5-Flash as the evaluator, further supporting its reliability and growing acceptance within the community.
> 2. Scalability: Conducting human evaluations for the scale of experiments required (thousands of samples across multiple dimensions and models) is prohibitively time-consuming and costly. We used UTMOS as a complementary objective metric for audio quality. We believe the combination of ALM-based semantic/style evaluation and UTMOS provides a rigorous and reproducible assessment standard.
>
> It is also important to emphasize that **both the base models and the SFT models are evaluated using the same Gemini-2.5-Flash judge**. This ensures that the reported improvements are internally consistent, directly comparable, and not artifacts of evaluator mismatch. From this perspective, the evaluation protocol is controlled and self-consistent across all baselines and our trained models, making the observed gains meaningful and persuasive.

---

> ### Author Response · Authors · 2025-11-25
> **Author Response (4/4)**
>
> ## **R4: Performance on External Benchmarks**
> ---
>
> **Yes**, we have conducted extensive evaluations on test sets that are completely independent of our generation pipeline. As also clarified in *Section 4.3*, *Section 4.4*, and in our earlier response [R2: Clarification of Generalization & Out-of-Domain Evaluation](https://openreview.net/forum?id=UrWdRcLINM&noteId=aXADO9Is76), our study includes multiple forms of OOD testing:
>
> 1. **URO-Bench (Table 5):** This benchmark evaluates general conversational abilities, including Oral Conversation, Understanding, and Reasoning, on a dataset entirely external to UltraVoice. On all three dimensions, the spoken dialogue models fine-tuned on UltraVoice show substantial improvements over their base versions and reach performance levels comparable to strong baselines such as Qwen2.5-Omni[4] and GLM4-Voice[5], demonstrating robust generalization to diverse conversational scenarios.
>
> 2. **EmoVoice-DB (Table 6):** In our controllable TTS validation experiment, we evaluate the model on EmoVoice-DB, a high-quality controllable TTS dataset that is fully independent of our generation pipeline. The controllable TTS model (UltraVoice-0.5B-SFT) trained on our synthesized data achieves strong performance on this external corpus, confirming that the fine-grained speech–style pairs produced by SOTA TTS systems are of high quality and transferable beyond the UltraVoice domain.
>
> These OOD experiments, spanning dialogue evaluation, reasoning-based tasks, and controllable TTS transfer, provide consistent evidence that models trained on UltraVoice generalize well beyond its distribution.
>
> ---
>
> We sincerely thank the reviewer for raising these questions. Your feedback allowed us to clarify our evaluation strategy and more clearly demonstrate the robustness and generalization ability of UltraVoice across diverse external benchmarks.
>
> ---
>
> [1] EmoVoice: LLM-based emotional text-to-speech model with freestyle text prompting.
> [2] Audio-aware large language models as judges for speaking styles.
> [3] URO-Bench: A comprehensive benchmark for end-to-end spoken dialogue models.
> [4] Qwen2.5-Omni Technical Report.
> [5] GLM-4-Voice: Towards Intelligent and Human-Like End-to-End Spoken Chatbot.

---

### Official Review · Reviewer_kfYh · 2025-10-29

**Soundness:** 3
**Presentation:** 3
**Contribution:** 2
**Rating:** 4
**Confidence:** 4

**Summary:**

UltraVoice addresses the lack of controllable expressivity in end-to-end spoken dialogue systems. The paper introduces a large, instruction-grounded dataset for fine-grained, multi-dimensional speech-style control across six dimensions—Emotion, Speed, Volume, Accent, Language, and Composite—comprising 100,770 dialogues (≈833 hours). The pipeline: (1) curate concise QA pairs from UltraChat; (2) inject diverse style instructions and generate style-aligned textual replies via GPT-4o; (3) synthesize speech pairs where user prompts use varied speakers/noise and system replies share a single fixed timbre for consistency (accents synthesized and normalized to the fixed voice); and (4) apply quality control with Whisper ASR, keeping clips <30 s and CER <20%.

To study downstream effects, the authors fine-tune representative voice agents (SLAM-Omni, VocalNet) spanning Qwen and LLaMA backbones. Evaluation covers all sub-controls with a 2.3k-item test set and reports Instruction-Following Rate (IFR) and MOS via an audio-language model, Gemini-2.5-Flash, plus objective metrics, like WER, emotion2vec similarity, and UTMOS. Fine-tuning on UltraVoice yields substantial gains: IFR +14.61–40.09 points and relative MOS +29–42%, with especially strong results for Emotion and Volume. General conversational ability also improves on URO-Bench (+10.84% Basic, +7.87% Pro), and the best SFT model achieves state-of-the-art among compared systems.

Notably, repurposing the replies as a controllable TTS corpus improves an EmoVoice model (WER 19.82 → 3.97 with MOS/IFR gains across Accent/Speed/Volume/Composite), indicating the dataset’s utility beyond dialogue.

**Strengths:**

1. UltraVoice includes six control dimensions (Emotion, Speed, Volume, Accent, Language, Composite), with instruction–response dialogue format; 100,770 dialogues (~833 hours). Authors positioned it as the first dataset explicitly designed for fine-grained, multi-dimension control in end-to-end voice agents.
2. By doing SFT on UltraVoice, fine-tuned models shows Instruction-Following Rate (IFR) improves by +14.61 to +40.09 percentage points across backbones and sizes; Mean Opinion Score (MOS) improves by ~29.1%–42.3%, relatively.
3. On URO-Bench, SFT models see overall gains on Understanding / Reasoning / Oral Conversation (avg. +10.84% Basic, +7.87% Pro); VocalNet-7B-SFT reaches SOTA among compared systems. Re-using the data for controllable TTS yields strong results (e.g., WER 19.82 → 3.97, plus MOS/IFR boosts on Accent/Speed/Volume/Composite), suggesting downstream utility beyond dialogue.

**Weaknesses:**

1. User prompts come from varied speakers/noise, but system replies use a single voice, and some Accent samples require voice conversion after TTS. This may cap speaker diversity and introduce voice conversion artifacts, creating a domain gap to real human recordings, and might influence downstream model's understanding of human voice features.
2. The Language dimension (Chinese/Korean/Japanese) is harder: authors claimed that LLaMA-based models show smaller gains or regressions in MOS/IFR there—likely due to limited multilingual pretraining exposure and/or limited multilingual SFT coverage. This is not rigorously proved. Further experiment or analysis are required to claim that.
3. All six stylistic dimensions use GPT-4o-generated style prompts (and style-conditioned responses). This centralizes the semantics of “emotion/speed/volume/accent/language” in one LLM’s worldview. This might bring bias to the synthesized dataset. It is crucial to evaluate   how large is the gap between the true distribution of human voice and the GPT-4o biased understanding of human voice.

**Questions:**

1.	In UltraVoice, the prompts for all six stylistic dimensions are generated by GPT-4o. I’m concerned that this might introduce some bias into the dataset due to GPT-4o’s own understanding of human emotions/speaking rate/accent, etc. If such bias exists, how can we quantify it so that others can try to balance this bias in future work?

2.	In the synthesized speech data, I listened to a few examples, and some are particularly impressive—such as Angry example 2. But I’m curious: across the whole dataset, what proportion of audio samples “might not make listeners feel it’s Angry”? Has there been any human evaluation of this proportion to verify that your filtering algorithm is robust enough to remove such data that could affect downstream models? I don't see any filter algorithm applied on this concern.

---

> ### Author Response · Authors · 2025-11-25
> **Author Response (1/4)**
>
> We sincerely appreciate your thorough review and insightful comments on UltraVoice. Your feedback highlights critical areas regarding data synthesis limitations and potential biases, which are crucial for advancing future spoken dialogue research. We address each weakness and question below.
>
> ---
>
> ## **R1: Clarification on Speaker Variety, Fixed Timbre Design, and Potential VC Artifacts**
>
> We appreciate the reviewer’s insightful observation on this point. **To clarify, UltraVoice intentionally includes varied speaker timbres and environmental conditions for the user-side utterances to enhance realism and diversify the dialogue scenarios.** In contrast, the agent’s timbre is fixed by design, as our goal is to train a spoken dialogue model with a consistent target voice rather than a multi-speaker TTS system. While part of the Accent subset involves voice conversion, objective evaluations confirm its reliability. Although part of the Accent subset involves voice conversion, objective evaluations support its reliability: the Accent dimension achieves a UTMOS of 4.08, higher than the overall average of 4.00, indicating that the VC process does not introduce noticeable artifacts or degrade naturalness. Therefore, this design does not impede the model’s ability to learn fine-grained style control.

---

> ### Author Response · Authors · 2025-11-25
> **Author Response (2/4)**
>
> ## **R2: Language Control Dimension Limitations**
> ---
>
> We appreciate the reviewer’s concern about the evidence supporting our explanation for the weaker improvements or regressions in the Language control dimension. **Our primary claim is that these weaker outcomes stem from model-dependent multilingual capacity rather than limitations of UltraVoice’s style labels or training methodology.** Empirically, LLaMA-based models (VocalNet-1B/8B) show slight MOS drops and largely flat IFR performance on Chinese/Korean/Japanese style control, whereas Qwen-based models (SLAM-Omni-0.5B, VocalNet-7B)—which have stronger multilingual (particularly Chinese) foundations—exhibit clear improvements. We hypothesize that these differences arise from variation in multilingual pretraining exposure and SFT coverage, together with the relatively limited size and diversity of our multilingual style data. We acknowledge that this does not constitute a rigorous causal proof; **fully validating it would require controlled multilingual pretraining and SFT ablations, which are beyond the scope of this work**. We will address these limitations and consider more comprehensive multilingual investigation in future work.

---

> ### Author Response · Authors · 2025-11-25
> **Author Response (3/4)**
>
> ## **R3: Quantifying Potential Bias from GPT-4o’s Worldview (W3 & Q1)**
> ---
>
> A central clarification we would like to make is that **while GPT-4o inevitably carries its own stylistic prior, our pipeline is explicitly designed to reduce over-dependence on any single phrasing or stylistic interpretation**, and the remaining bias is fundamentally difficult to quantify due to the lack of fine-grained human ground-truth style distributions.
>
> - **Mitigation in Synthesis**: To mitigate over-centralization, **our data generation pipeline mandates the use of synonyms or alternative expressions for styles** (e.g., using "joyful tone," "cheerful voice," or "ecstatic tone" rather than just a fixed category). This design choice promotes linguistic diversity and forces the model to engage with varied semantic expressions of a style, making the instruction following more robust and less reliant on a single phrasing.
> - **Quantifying Bias (The Challenge)**: Quantifying the exact "gap between the true distribution of human voice and the GPT-4o biased understanding" is challenging because reliable, fine-grained human ground truth distribution for expressive dialogue styles is currently lacking. We can only quantify the alignment of the LLM's judgment with human perception. Previous studies[1] found that ALM judges (like Gemini-2.5) demonstrate an agreement with human judges on speaking style evaluation (Pearson’s r of 0.640) that is comparable to human-human agreement. This suggests the LLM's perception of style is generally aligned with human perception. **We agree that developing novel, fine-grained evaluation metrics that accurately measure the fidelity of synthetic style distribution against real human distributions is an urgent need for future research.**

---

> ### Author Response · Authors · 2025-11-25
> **Author Response (4/4)**
>
> ## **R4: Verification of Emotional Quality and Robustness of Filtering (Q2)**
> ---
>
> We thank the reviewer for this question. In the current version, we indeed do not apply an explicit emotion-consistency filter for samples that may “not really sound Angry,” nor do we report a human-estimated proportion of such cases. There are two main reasons:
> - At the present data scale, **utterance-level human emotion annotation would be prohibitively costly**.
> - Existing automatic emotion models (e.g., Emo2Vec-style encoders) are still noticeably less reliable than human listeners for fine-grained conversational emotions, so using them as a hard filter risks discarding many valid samples and amplifying model bias.
>
> Given these limitations, we adopt a scalable and conservative strategy: synthesising emotional speech with strong TTS systems such as GPT-4o-audio-preview, applying objective filters (CER < 20%, duration constraints), **and validating dataset usefulness through substantial downstream improvements in emotional and stylistic control across multiple spoken dialogue models and controllable TTS systems.** These empirical results indicate that the emotional supervision contained in UltraVoice is sufficiently robust for practical learning. Developing more reliable emotion-consistency filtering—integrating stronger emotion models with sampled human validation—remains a promising future work and is an active research challenge for the broader community.
>
> ---
>
> In summary, we hope that the above clarifications and additional discussions adequately address your concerns regarding UltraVoice, and we would like to once again sincerely thank you for your careful review and constructive feedback.
>
> ---
>
> [1] Audio-aware large language models as judges for speaking styles.

---

### Official Review · Reviewer_215S · 2025-11-01

**Soundness:** 3
**Presentation:** 3
**Contribution:** 3
**Rating:** 6
**Confidence:** 3

**Summary:**

This paper introduces UltraVoice, a large speech dialogue dataset designed to give spoken dialogue models fine-grained control over how they speak across emotion, speed, volume, accent, language, and composite styles. The authors build the corpus through a four-stage pipeline that curates text, injects style instructions, synthesizes stylized speech, and filters for quality, then uses it to fine-tune popular end-to-end dialogue models. The fine-tuned models show stronger adherence to style instructions while keeping or improving general conversational ability on an external benchmark. The dataset also transfers to controllable text-to-speech training, indicating broad utility for expressive and natural voice generation.

**Strengths:**

- The paper targets an important gap by enabling fine-grained style control across six dimensions and provides the first large-scale dialogue dataset designed for this purpose.
- The dataset construction pipeline is thorough, with explicit filtering via CER and audio duration for quality enhancement.
- Fine-tuning on UltraVoice raises instruction following and subjective naturalness across models and across most style dimensions. General conversational ability also improves on URO-Bench.

**Weaknesses:**

- The corpus is fully synthetic, built with GPT-4o and several TTS or VC systems, which may introduce artifacts and reduce diversity.
- Some evaluations are done by an ALM judge, which raises concerns about reproducibility and potential bias in subjective scores. A stronger human evaluation component would help.
- Since the authors argue that existing controllable TTS data are not suitable for fine-tuning dialogue models, the paper should add comparative fine-tuning results using those datasets. This would substantiate the claim.
- Accent control uses Edge TTS with post-hoc voice conversion to a fixed timbre, which may introduce artifacts or distortion. An analysis should verify its quality and check whether other styles, such as speed or emotion, remain unchanged.

**Questions:**

Please refer to my comments in the weaknesses section.
I also noticed two small typos:
- In Sec. 4.1, add a space after "(Table 14)".
- In Appendix E.8, "tyle_description" should be "style_description".

---

> ### Author Response · Authors · 2025-11-25
> **Author Response (1/4)**
>
> We sincerely thank you for your careful review and constructive feedback. We also appreciate your recognition of the effort that went into developing UltraVoice.
>
> ---
>
> ## **R1: Concerns Regarding Synthetic Corpus Quality and Diversity**
> ---
>
> Thank you for raising concerns regarding the quality of our synthetic corpus. UltraVoice was designed with scalability and consistency in mind. **Given the dataset’s large size, we adopted CER and duration constraints as the core filtering criteria, which is a practical and widely used strategy for large-scale synthetic speech datasets.**[2]
>
> Across all stylistic dimensions, we employed state-of-the-art TTS systems available at the time (e.g., GPT-4o-audio-preview, CosyVoice), ensuring high fidelity and reliable style rendering.
>
> **Most importantly, the dataset’s quality and usefulness are empirically validated:**
> - Both in-domain and out-of-domain evaluations (URO-Bench and EmoVoice-DB) show substantial improvements in fine-grained style control and overall spoken dialogue abilities.
> - Additionally, training a controllable TTS model on UltraVoice confirms that the synthesized style–speech pairs are sufficiently accurate and natural for reliable style learning.
>
> These results collectively demonstrate that our filtering pipeline—despite being based on CER and duration—effectively preserves data quality while maintaining scalability, enabling UltraVoice to serve as a strong resource for training expressive spoken dialogue models.

---

> ### Author Response · Authors · 2025-11-25
> **Author Response (2/4)**
>
> ## **R2: Justification for Using Audio-Language Models (ALM) as Judges**
> ---
>
> We recognize the preference for human evaluation, which is typically the most accurate way to assess subjective qualities. However, human evaluation is "time-consuming and costly". **We argue that employing an ALM judge is a reliable and scalable alternative given the strong correlation with human perception, while simultaneously using comprehensive objective metrics.**
>
> - Alignment with Established Methodology: Our evaluation paradigm, which uses Gemini-2.5-Flash, follows similar methodologies utilized in contemporary work. [1][2]
> - High Agreement with Human Judges: The choice of Gemini is motivated by findings that advanced ALMs demonstrate high consistency with human judgments[3]. Specifically, independent research on evaluating speaking styles (Voice Style Instruction Following) found that the average Pearson’s r correlation between the Gemini judge and human evaluators was 0.640, which was higher than the average pairwise human-human correlation of 0.596. This result strongly validates the effectiveness of using Gemini as a judge for assessing speaking styles.

---

> ### Author Response · Authors · 2025-11-25
> **Author Response (3/4)**
>
> ## **R3: Clarification — UltraVoice Trains a Dialogue Model, Not a Standalone TTS System**
> ---
> Thank you for the suggestion. We would like to clarify a key point that may have caused misunderstanding. **UltraVoice is developed to train and evaluate an end-to-end spoken dialogue model, rather than a standalone controllable TTS system.** The model operates in an instruction–response setting, where style control is embedded within conversational semantics.
>
> Existing controllable TTS datasets are **not suitable** for this objective. They consist of isolated sentences without instruction-driven context, and therefore cannot meaningfully evaluate a dialogue model’s ability to interpret style directives within a conversational framework. Converting such datasets into “dialogue-like” format would reduce the task to plain sentence reading, which does not reflect the intended model capability.

---

> ### Author Response · Authors · 2025-11-25
> **Author Response (4/4)**
>
> ## **R4: Analysis of Accent Control Quality and Style Consistency**
> ---
>
> Thank you for the question. The Accent dimension indeed uses a hybrid Edge TTS + CosyVoice-300M VC pipeline, whose sole purpose is to enforce a fixed timbre for system responses. **We verified its quality through our standard filtering: all samples passed the CER <20% threshold, and the Accent subset achieved a UTMOS of 4.08, higher than the overall dataset average (4.00) and the strongest among all six style dimensions.** This indicates that the VC step did not introduce noticeable artifacts.
>
> Moreover, if VC had introduced harmful distortions, it would have impaired the model’s learning of other style dimensions. **Instead**, fine-tuning on UltraVoice led to substantial gains across all other styles—including Emotion, Speed, and Volume (e.g., +54.13%, +24.06%, +28.39% for VocalNet-1B). This confirms that the Accent data does not degrade stylistic consistency and in fact contributes positively to robust multi-dimensional control.
>
> ---
>
> We once again thank the reviewer for the thorough evaluation and constructive suggestions, which have helped us further strengthen the clarity and validity of UltraVoice. We also appreciate the reviewer for pointing out the typos, and we will update them in the new version.
>
> ---
>
> [1] URO-Bench: A comprehensive benchmark for end-to-end spoken dialogue models.
> [2] EmoVoice: LLM-based emotional text-to-speech model with freestyle text prompting.
> [3] Audio-aware large language models as judges for speaking styles.

---

### Author Response · Authors · 2025-12-03
**Rebuttal Reviews**

We sincerely thank the reviewers for their thoughtful feedback and appreciate the AC’s precious time and patience.

Below is a summary of our rebuttal.

---

## **S1.Core Contribution**
- **First Large-Scale Multi-Dimensional Style-Controlled Speech Dialogue Dataset**: We present the first large-scale speech dialogue dataset explicitly designed for fine-grained multi-dimensional speech style control, filling a major gap in the field. The dataset supports six dimensions of style manipulation and features a comprehensive, reproducible construction pipeline.
  - acknowledged by reviewers `215S`, `kfYh`, `Mx6y`, `hgwR`.
- **Comprehensive Model Evaluation Experiments**: Using this dataset, we conduct extensive SFT experiments on mainstream end-to-end spoken dialogue models, including SLAM-Omni and the VocalNet series. Results show that the dataset significantly enhances style controllability and expressiveness. Furthermore, on the out-of-domain benchmark URO-Bench, models fine-tuned with UltraVoice achieve consistent and substantial performance gains.
  - acknowledged by reviewers `215S`, `kfYh`, `Mx6y`, `hgwR`.
- **Independent Verification of Style Quality**: To further validate the style quality of our dataset, we fine-tune a pre-trained TTS model using synthesized audio generated under different style controls. The resulting TTS model demonstrates stable and controllable multi-dimensional style generation, providing independent evidence of the dataset’s reliability and practical utility.
  - acknowledged by reviewers `kfYh`, `hgwR`.

Our experiments demonstrate that UltraVoice significantly enhances the fine-grained, multi-dimensional speaking-style controllability of speech dialogue models. We hope this high-quality dataset will, like VoiceAssistant400K and InstructS2S, be widely adopted, driving advances in speech dialogue models and expanding the limits of model capabilities.

---

## **S2.Main concerns raised by reviewers**
- **Using only synthetic data to address fine-grained style-control challenges is viewed as lacking novelty.**
  - Our contributions are clearly summarized in the `S1.Core Contribution`, which are also acknowledged by the reviewers.
  - Given the high cost and scalability issues of real-speech collection, we use state-of-the-art TTS models to generate style-labeled data with reliable quality. Many recent works follow the same paradigm to improve spoken dialogue models (e.g., `Llama-Omni ICLR2025`, `SLAM-Omni ACL2025`, `EmoVoice MM2025`, `VocalNet EMNLP2025`).
  - SFT and both in-domain and out-of-domain evaluations show that our dataset effectively enhances style controllability.
- **Concerns regarding the evaluation setup and reliability of ALM-based style assessment.**
  - The same ALM model is used across all experiments. This ensures that all reported improvements are internally consistent and not influenced by discrepancies across evaluators. From this perspective, the relative gains observed in our experiments are fully self-consistent and therefore meaningful and persuasive.
  - Modern ALM systems (e.g., Gemini 2.5) have demonstrated strong multimodal understanding correlates well with human judgments, and many contemporary studies adopt them for large-scale evaluation, making our setup reasonable.

---

## **S3. Clarification For Some Misunderstanding**

We believe reviewers `Mx6y` and `hgwR` may have misunderstood certain aspects of the paper. We provide the following clarifications:

> `Mx6y`: Lack of Generalization & OOD Evaluation

Response: Our evaluation **does include multiple OOD tests**, as shown in Sections 4.3 and 4.4, including URO-Bench and transfer evaluation on the external real-speech dataset EmoVoice-DB. The model maintains consistent performance across these settings, demonstrating solid out-of-distribution generalization.

> `hgwR`: The experiment in Section 4.4 is inappropriate

Response: To clarify, the experiment in Section 4.4 is not intended as a benchmark comparison, nor does it attempt to match or surpass SOTA performance on EmoVoice-DB. **Its sole purpose is to validate the usability and quality of UltraVoice’s multi-dimensional style-control data.** We fine-tune the controllable TTS model (UltraVoice-0.5B-SFT) using only English synthetic style dimensions (Emotion, Speed, Volume, Accent, Composite), ensuring that WER is computed strictly on English data and free from cross-lingual confounding factors.

We place high importance on the accuracy and fairness of the review process.

Given these clarifications, we respectfully ask the AC and program chairs to **revisit the assessments from `Mx6y` and `hgwR`** and **adjust their influence in the final decision accordingly**. Our goal is simply to ensure evaluations are based on accurate understanding and consistent criteria.

---
We appreciate the reviewers and the AC for their hard work and their commitment to maintaining a high-quality review process.

Best regards,

All authors

---

### Note · Authors · 2026-01-05

I have read and agree with the venue's withdrawal policy on behalf of myself and my co-authors.